# Hierarchical Multi-Scale Molecular Conformer Generation

**Jiapeng Hu**[1]    **Weizhi Gao**[1]    **Zhichao Hou**[1]    **Xiaorui Liu**[1]
[1]North Carolina State University
jhu48@ncsu.edu.cn, wgao23@ncsu.edu.cn, zhou4@ncsu.edu, xliu96@ncsu.edu

## Abstract

Molecular conformer generation is a fundamental task for drug discovery and material design. Although deep generative models have progressed in this area, existing methods often overlook the hierarchical structural organization inherent to molecules, leading to poor-quality generated conformers. To address this challenge, we demonstrate that capturing the spatial arrangement of key substructures, such as scaffolds, is essential, as they serve as anchors that define the overall molecular distribution. In this paper, we propose a hierarchical multi-scale molecular conformer generation framework (MSGEN), designed to enhance key substructure awareness by leveraging spatially informed guidance. Our framework initiates the generation process from coarse-grained key substructures, progressively refining the conformer by utilizing these coarser-scale structures as conditional guidance for subsequent finer-scale stages. To bridge scale discrepancies between stages, we introduce a molecular upsampling technique that aligns the structural scales, ensuring smooth propagation of geometric guidance. Extensive experiments on standard benchmarks demonstrate that our framework integrates seamlessly with a wide range of existing molecular generative models and consistently generates more stable and chemically plausible molecular conformers.

## 1 Introduction

Molecular conformer generation is a fundamental task in computational chemistry with applications in drug discovery, material design, and property prediction, where we predict the 3D geometry of a molecule from its 2D graph representation (Guimarães et al., 2012; Schütt et al., 2021; Axelrod & Gomez-Bombarelli, 2023). Recent advances in deep generative models, such as diffusion models (Dhariwal & Nichol, 2021) and flow matching (Lipman et al., 2022), have demonstrated strong potential for generating accurate and diverse molecular conformers (Simm & Hernández-Lobato, 2019; Shi et al., 2021; Jing et al., 2022). However, many existing methods focus on generating atomic coordinates or local geometries, such as torsion angles, while neglecting higher-level structural information. As a result, they often fail to preserve essential substructures, such as ring systems, leading to chemically implausible conformers (Zhou et al., 2023; Fan et al., 2024).

To improve the chemical validity of conformer generation, existing work incorporated different structural guidance. These efforts can be categorized by the structural granularity they emphasize. Methods like Subgdiff (Zhang et al., 2024), which preserve local structures by denoising within subgraphs, often disrupt global consistency. Fragment-level approaches aim to retain

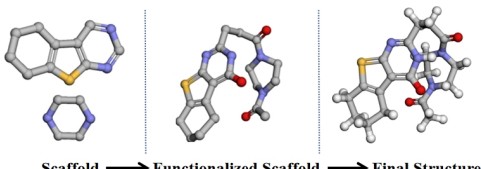

Figure 1: Multi-Scale Molecule Representation.

coarse molecular features. Early design (Qiang et al., 2023) captures fragment information at a single resolution but lacks a coarse-to-fine generative process. Subsequent work (Park & Shen, 2024) adopts a coarse-to-fine strategy based on equivariant blurring diffusion (Rissanen et al., 2022). Nevertheless, these fragment-based methods generally treat all fragments uniformly, overlooking differences in functional importance and conformational flexibility. Therefore, it is essential to effectively organize and utilize structural information in generative models.

As shown in Figure 1, molecular structures naturally exhibit a hierarchical organization (Zhang et al., 2012; Gennis, 2013). In chemistry, this hierarchical nature reflects how molecules are assembled from basic structural motifs, such as rigid rings and flexible linkers, into complex functional architectures (Boninsegna et al., 2018). Consequently, generating valid molecular conformers requires more than just focusing on a single structural scale. Despite recent efforts to incorporate coarse-grained structural features (Qiang et al., 2023; Park & Shen, 2024), they often fail to capture the hierarchical assembly process that governs spatial dependencies across structural scales. In molecular systems, the arrangement of fine-grained components is typically conditioned on the positioning of key coarser substructures. In drug-like molecules, global scaffolds often constrain the spatial arrangement of flexible side chains, thereby ensuring chemically plausible geometries. This structural dependency motivates the incorporation of geometric guidance as an inductive bias to promote global consistency and chemical validity in conformer generation. To evaluate the effectiveness and importance of structural guidance, we conducted a preliminary study comparing several generative models that incorporate structural features with a variant that leverages scaffold positional information in Section 3. *The results demonstrate the strong power of substructure positions to enhance structural awareness, preserving geometric consistency and chemical validity during generation.*

However, the geometric guidance is usually not available during generation. Based on these insights, we propose MSGEN, a general framework for molecular conformer generation that preserves chemical properties while supporting the multi-scale nature of molecular structures. Our framework adopts a multi-scale generation process, progressively refining molecular structures from coarse to fine scale. The output of each stage serves as a flexible and detailed structural guidance for the subsequent stage. To mitigate the structural scale mismatch between coarse and fine representations, we introduce molecular upsampling. It can bridge the structural gap between stages, enabling more efficient and chemically valid propagation of structural guidance. To validate our framework, we conducted comprehensive experiments on standard molecular conformer benchmarks, including conformer generation and property prediction tasks. The results showed that the MSGEN framework consistently enhances existing baseline models in generation quality, leading to more chemically plausible conformers. In summary, our contributions are fourfold:

- We conducted a preliminary study to compare different structural guidance strategies to demonstrate the strong power of substructure positions in enhancing structural awareness.

- We proposed MSGEN, a general framework for molecular conformer generation that explicitly reflects the hierarchical nature of molecular structures. This framework adopts a multi-scale generation process from coarse scaffolds to full-atom details.

- We introduce molecular upsampling to address the structural scale mismatch between coarse and fine representations, facilitating the efficient and chemically consistent propagation of structural guidance across stages.

- We perform extensive experiments and ablation studies to assess the effectiveness of our framework in enhancing the structural awareness of baseline models. The improved awareness leads to the generation of more stable and chemically plausible conformers.

## 2    RELATED WORK

**Hierarchical Diffusion Model.** Diffusion models (Ho et al., 2020; Song et al., 2020; Su et al., 2024) generate data by iteratively denoising samples from a prior distribution and have achieved notable success in vision tasks. To enhance the efficiency and structural fidelity of generative processes, hierarchical diffusion models (Ho et al., 2022a; Saharia et al., 2022b; Ho et al., 2022b; Wang et al., 2024) decompose the generation task into multiple stages, progressively refining coarse structures into fine details. Motivated by the inherent hierarchical organization of molecules, recent works on molecular graph generation (Jin et al., 2020), protein structure (Yang & Gómez-Bombarelli, 2023) have shown the effectiveness of leveraging this multi-scale nature. In recent molecular conformer generation, EBD (Park & Shen, 2024) leverages a blurring diffusion (Rissanen et al., 2022) to recover full atom details from fragments, but it lacks multi-scale structure awareness, as it treats all fragments uniformly. It remains unclear how to design hierarchical diffusion models for molecular conformer generation, and this work presents the first study to address this challenge.

**Molecular Conformer Generation.** Various machine learning based approaches have been proposed for molecular conformer generation (Mansimov et al., 2019; Simm & Hernández-Lobato, 2019; Xu et al., 2021a; Shi et al., 2021; Ganea et al., 2021). Recently, the diffusion-based method (Xu et al., 2022; Jing et al., 2022; Wang et al., 2023) has been introduced to this domain, showing great promise in generating diverse and accurate conformers. GeoDiff (Xu et al., 2022) introduced a diffusion model that learns to denoise 3D atomic positions conditioned on molecular graphs. TorsionDiff (Jing et al., 2022) designs a diffusion model on the torsion angles. MCF (Wang et al., 2023) trained a diffusion model over functions that map elements from the molecular graph to points in 3D space. However, most existing methods primarily focus on a single scale during molecular conformer generation, neglecting the inherent multi-scale organizations of molecules.

**Equivariant Deep Learning.** Recent work in AI for Science has focused on incorporating physical and chemical priors, particularly the rotational symmetries captured by the Euclidean group $SO(3)$. In the past few years, various invariant and equivariant architectures have been developed for Cartesian (Schütt et al., 2017; Gasteiger et al., 2020; Satorras et al., 2021; Liao & Smidt, 2022; Thölke & De Fabritiis, 2022; Wu et al., 2023; Chen et al., 2024) and spherical basis (Thomas et al., 2018; Fuchs et al., 2020; Brandstetter et al., 2021; Frank et al., 2022). For molecular generation, initial methods like ConfGF Shi et al. (2021) utilize invariant networks for inter-atomic distances, whereas the equivariant Geometric GNNs have been used in GeoDiff and ET-Flow (Hassan et al., 2024).

# 3 Preliminary Study on Molecule Structural Guidance

**Problem Definition and Notations.** Molecular conformer generation aims to produce chemically valid and novel molecular coordinates $\mathbf{R}$, conditioned on a given molecular graph $\mathcal{G}$. In this paper, we denote each molecule with $n$ atoms as an undirected graph $\mathcal{G} = (\mathcal{V}, \mathcal{E}, \mathbf{X})$, where $\mathcal{V} = \{v_i\}_{i=1}^n$ represents atoms, $\mathcal{E} = \{e_{ij} \mid (i,j) \subseteq |\mathcal{V}| \times |\mathcal{V}|\}$ represents inter-atomic bonds, and $\mathbf{X}$ encodes node features, e.g. the atom type. Each atom $v_i$ is associated with a coordinate $\mathbf{r}_i \in \mathbb{R}^3$, and the molecular conformer can be represented as a matrix $\mathbf{R} = [\mathbf{r}_1, \ldots, \mathbf{r}_n] \in \mathbb{R}^{n \times 3}$.

**Limitations of Existing Works.** Since vanilla diffusion-based approaches (Xu et al., 2022) fail to preserve essential chemical substructures during generation, several extensions (Zhang et al., 2012; Park & Shen, 2024) attempt to preserve key structures either through subgraph denoising or fragment-level features. These methods improve local connectivity and fragment consistency, but they neglect the dominant role of key structural components such as ring systems and heavy-atom backbones (Velkoborsky & Hoksza, 2016; Manelfi et al., 2021). Without explicitly constraining these key substructures, generated conformers may satisfy local validity but still deviate from realistic global arrangements. Therefore, we propose that incorporating geometric structural guidance, such as **the positions of key substructures**, can serve as a more chemically meaningful prior. These will help anchor the spatial arrangement of essential structural motifs. By integrating such information, *we aim to enhance structural awareness and reduce uncertainty in the generation*

To validate our hypothesis, we designed a preliminary study to evaluate model performance under four types of guidance. Specifically, we conduct a controlled experiment with (1) no structural guidance (Xu et al., 2022), (2) subgraph-level structural guidance (Zhang et al., 2024), which enforces local connectivity, (3) fragment-level guidance (Park & Shen, 2024), which emphasizes chemical motif identity, and (4) geometric structural guidance, which masks the positions of light atoms while retaining the ground-truth positions of the heavy-atom backbone. All of those results use 500 diffusion time steps during generation.

Table 1: Geometric Evaluation on GEOM-Drugs ($\delta = 1.25\text{Å}$). The results show that the geometric structural guidance effectively enhances the generation quality.

| Guidance Type | COV-R (%) ↑ | | MAT-R (Å) ↓ | | COV-P (%) ↑ | | MAT-P (Å) ↓ | |
|---|---|---|---|---|---|---|---|---|
| | Mean | Median | Mean | Median | Mean | Median | Mean | Median |
| None (Xu et al., 2022) | 64.12 | 75.56 | 1.1444 | 1.1246 | 43.16 | 42.02 | 1.3806 | 1.3314 |
| Local (Zhang et al., 2024) | 74.30 | 86.43 | 1.0003 | 0.9905 | −− | −− | −− | −− |
| Fragment (Park & Shen, 2024) | 84.88 | 94.97 | 0.9118 | 0.8788 | 64.74 | 68.19 | 1.1465 | 1.0876 |
| Geometric(ours) | **99.58** | **100.00** | **0.5035** | **0.4899** | **88.56** | **94.70** | **0.7816** | **0.7515** |

As shown in Table 1, incorporating geometric position guidance yields substantial improvements across all evaluation metrics compared to the other approaches, indicating that geometric structural guidance enables the model to capture substructures and their global arrangements more effectively than existing approaches. More implementation details are in Appendix C.4.

# 4 HIERARCHICAL MULTI-SCALE MOLECULAR CONFORMER GENERATION

In this section, we elaborate on our novel framework MSGEN, designed to enhance structural awareness during conformer generation while preserving chemical properties. The motivation for a multi-scale design arises from the inherently hierarchical nature of molecules (Manelfi et al., 2021). Our framework adopts a hierarchical molecular conformer generation process, where each stage incrementally refines the molecular structures based on the geometric structural guidance provided by the previous stage. We further introduce molecular upsampling to reconcile the mismatch of structural scale across stages, enabling effective and efficient guidance propagation.

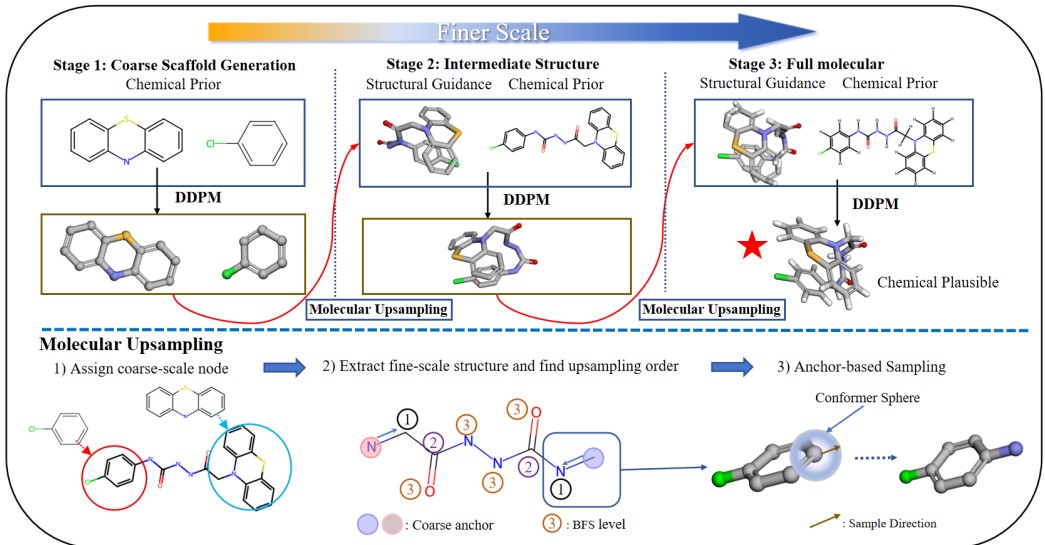

Figure 2: Illustration of the MSGEN framework with three stages. (a) Each stage generates a refined representation of the molecular conformer, guided by previous stages, with the outputs connected via molecular upsampling. (b) In molecular upsampling, atoms in the fine-scale representation are first ordered and then sequentially assigned coordinates via anchor-based sampling.

## 4.1 MULTI-SCALE MOLECULAR CONFORMER GENERATION

The results in Section 3 motivate the use of geometric structural guidance during generation. However, such guidance is not directly accessible when only the molecular graph is available at inference time. To address this limitation, we propose multi-scale molecular conformer generation, where each fine-scale stage is conditioned on the structural representations generated at the previous coarse-scale stage. We define a $K$-level hierarchical representation of molecular structure based on its inherent multi-scale organization, as shown in Figure 1, which preserves chemical substructures across different levels of granularity. Each level corresponds to a subgraph $\mathcal{G}^k$, forming a nested sequence $\mathcal{G}^1 \subset \mathcal{G}^2 \subset \cdots \subset \mathcal{G}^K = \mathcal{G}$ from coarse to fine. The atomic coordinates associated with $\mathcal{G}^k$ are denoted as $\mathbf{R}^k \in \mathbb{R}^{n_k \times 3}$, where $n_k$ is the number of atoms in $\mathcal{G}^k$. We model the distribution $p_{\theta_1}(\mathbf{R}^1|\mathcal{G}^1)$ at the first stage and model $p_{\theta_k}(\mathbf{R}^k|\mathcal{G}^k, \mathbf{R}^{k-1})$ for subsequent stages using individual generative models parameterized with $\theta_k$, where $k$ is the index of stages. While we use DDPM (Ho et al., 2020) as an illustrative example, our framework is compatible with a variety of generative models, including flow matching (Lipman et al., 2022).

**Forward Process.** For each stage $k = 1, \cdots, K$, the forward process is defined as follows: $q_k(\mathbf{R}_t^k|\mathbf{R}_0^k) = \mathcal{N}(\mathbf{R}_t^k; \sqrt{\bar{\alpha}_t^k}\mathbf{R}^k, (1 - \bar{\alpha}_t^k)\mathbf{I})$, where $t \in \{1, ..., T_k\}$ is the diffusion step and $\bar{\alpha}_t^k$ denotes the cumulative noise schedule for stage $k$. Note that we employ separate forward processes for each stage, allowing our framework the flexibility to construct stage-specific training data aligned with the corresponding distributions.

**Reverse Process.** The reverse process generates conformers in a guided, hierarchical manner, progressively transitioning from coarse structural scaffolds to the complete conformer. At the first stage, the model generates the coarse-scale structure $\mathbf{R}^1$ without guidance. The denoising process at this stage is formulated as: $p_{\theta_1}(\mathbf{R}_{t-1}^1|\mathcal{G}^1, \mathbf{R}_t^1) = \mathcal{N}(\mathbf{R}_{t-1}^1; \mu_{\theta_1}(\mathcal{G}^1, \mathbf{R}_t^1, t), (1 - \alpha_t^1)\mathbf{I})$, where $\mu_{\theta_1}(\cdot)$

is a equivariant neural network parameterized by $\theta_1$ and $\alpha_t^1$ is the noise schedule. For subsequent stages $k = 2, ..., K$, the reverse process progressively refines the structure to $\mathbf{R}^k$, leveraging the coarse representation from the previous stage. The conditional denoising process is formulated as: $p_{\theta_k}(\mathbf{R}_{t-1}^k | \mathcal{G}^k, \mathbf{R}_t^k, \mathbf{R}_{\text{cond}}^k) = \mathcal{N}(\mathbf{R}_{t-1}^k; \mu_{\theta_k}(\mathcal{G}^k, \mathbf{R}_t^k, \mathbf{R}_{\text{cond}}^k, t), (1 - \alpha_t^k)\mathbf{I})$, where $\mu_{\theta_k}(\cdot)$ and $\alpha_t^k$ are the equivariant neural network and the noise schedule for stage $k$. $\mathbf{R}_{\text{cond}}^k$ is a geometric structural guidance derived from the coordinates of the previous level $\mathbf{R}^{k-1}$: $\mathbf{R}_{\text{cond}}^k = \mathcal{F}_k(\mathbf{R}^{k-1}, \mathcal{G}^{k-1}, \mathcal{G}^k)$, where $\mathcal{F}_k(\cdot)$ is a conditioning function. During sampling, we iteratively compute $\mathbf{R}_{\text{cond}}^k$ based on the output sampled from the previous stage $\hat{\mathbf{R}}^{k-1}$, effectively capturing and leveraging the multi-scale hierarchical structure of molecular conformers.

---

**Algorithm 1:** Topological Sort

**Input:** Molecular graph $\mathcal{G} = (\mathcal{V}, \mathcal{E})$,
        Atom sets $\mathcal{I}_{\text{coarse}}, \mathcal{I}_{\text{fine}}$
**Output:** Upsampling Order $\mathcal{O}$
$\mathcal{R} = \mathcal{I}_{\text{fine}} \setminus \mathcal{I}_{\text{coarse}}$ ;
$\mathcal{B} = \{i | i \in \mathcal{R}, \exists j \in \mathcal{I}_{\text{coarse}}, (i, j) \in \mathcal{E}\}$ ;
Initialize visited set $\mathcal{V} \leftarrow \mathcal{B}$ ;
Initialize queue $\mathcal{Q} \leftarrow \mathcal{B}$ ;
**while** $\mathcal{Q}$ *not empty* **do**
     Pop $i$ from $\mathcal{Q}$ and append to $\mathcal{O}$ ;
     **for** *each* $j \in \mathcal{N}(i)$ **do**
         **if** $j \notin \mathcal{V}$ *and* $j \notin \mathcal{I}_{\text{coarse}}$ **then**
             **if** *Any* $k \in \mathcal{N}(j)$ *in* $\mathcal{V}$ **then**
                Add $j$ into $\mathcal{O}$ and $\mathcal{V}$
             **end**
         **end**
     **end**
**end**
Append any remaining atoms in $\mathcal{R}$ to $\mathcal{O}$ ;
**return** $\mathcal{O}$

---

**Algorithm 2:** Anchor-based Upsampling

**Input:** Coarse-level positions $\mathbf{R}_{\text{coarse}}$, Atom
        set $\mathcal{I}_{\text{coarse}}$, Molecular graph $\mathcal{G}$,
        Upsampling order $\mathcal{O}$, Threshold $\tau$,
**Output:** Fine-level conditions $\mathbf{R}_{\text{cond}}$
Assign positions: $\mathbf{R}_{\text{cond}}[\mathcal{I}_{\text{coarse}}] \leftarrow \mathbf{R}_{\text{coarse}}$;
Compute coarse centroid $\mathbf{c} \leftarrow \text{mean}(\mathbf{R}_{\text{coarse}})$;
Initialize positioned set $\mathcal{P} \leftarrow \mathcal{I}_{\text{coarse}}$ ;
**for** *each atom* $i \in \mathcal{O}$ **do**
     $\mathcal{N}(i) \leftarrow \mathcal{N}(i) \cap \mathcal{P}$ ;
     **if** $\mathcal{N}_i \neq \emptyset$ **then**
         Anchor $\mathbf{a}_i \leftarrow \frac{1}{|\mathcal{N}_i|} \sum_{j \in \mathcal{N}_i} \mathbf{R}_j$;
     **else**
         Anchor $\mathbf{a}_i \leftarrow \mathbf{c}$;
     **end**
     Sample direction $\mathbf{d}_i \sim \mathcal{N}(\mathbf{0}, \mathbf{I})$ ;
     $\mathbf{R}_{\text{cond}}[i] \leftarrow \mathbf{a}_i + \tau \cdot \frac{\mathbf{d}_i}{||\mathbf{d}_i||}$;
     Add $i$ into $\mathcal{P}$
**end**
**return** $\mathbf{R}_{\text{cond}}$

---

## 4.2 MOLECULAR UPSAMPLING

In our multi-scale framework, coarse-level outputs provide essential context for finer resolutions. However, directly using $\mathbf{R}^{k-1}$ as the guidance $\mathbf{R}_{\text{cond}}^k$ can lead to scale mismatch, hindering effective information transfer. While upsampling methods like interpolation or transposed convolutions are common in vision tasks (Ronneberger et al., 2015; Saharia et al., 2022a), they are not well-suited to molecular conformers due to the continuous nature of 3D molecular structures. To address this, we introduce a molecular upsampling strategy that respects chemical graph connectivity.

Formally, given coarse-level atoms $\mathcal{I}_{\text{coarse}}$ with positions $\mathbf{R}_{\text{coarse}} \in \mathbb{R}^{m \times 3}$ and the full molecular graph $\mathcal{G}$, our goal is to infer a chemically meaningful prior $\mathbf{R}_{\text{cond}} \in \mathbb{R}^{n \times 3}$ for atoms in the fine set $\mathcal{I}_{\text{fine}}$. Since atoms are arranged in a non-Euclidean molecular graph $\mathcal{G}$, some fine atoms rely on the positions of already positioned atoms to achieve spatial consistency. To ensure valid ordering and avoid cycles, we perform a topological sort over $\mathcal{G}$, yielding an upsampling sequence $\mathcal{O}$ (see Algorithm 1). Following this order, we iteratively assign positions to fine-level atoms using their placed neighbors as geometric anchors (see Algorithm 2). To systematically track the atoms whose positions have been determined, we maintain a positioned set $\mathcal{P}$, initialized with all coarse-level atoms. For each fine atom $i \in \mathcal{O}$, we identify its positioned neighbors and compute:

$$\mathbf{R}_{\text{cond}}[i] = \frac{1}{\mathcal{N}(i) \cap \mathcal{P}} \sum_{j \in \mathcal{N}(i) \cap \mathcal{P}} \mathbf{R}_j + \tau \cdot \mathbf{d}_i, \quad \mathbf{d}_i \sim \mathcal{N}(0, I), \qquad (1)$$

where $\tau$ controls sampling radius and $\mathbf{d}_i$ introduces spatial variation while preserving structural continuity. The atom is then added to $\mathcal{P}$ for subsequent steps. This process ensures that molecular geometry is constructed progressively, respecting both chemical connectivity and spatial coherence.

### 4.3 MODEL TRAINING

In this section, we introduce our conditional augmentation to avoid the distribution shift problem and elaborate on the loss function for multi-stage training. We use DDPM following Section 4.1.

**Conditional Augmentation.** In standard diffusion training, the forward process uses the ground-truth $\mathbf{R}^{k-1}$ to construct training data, while inference relies on the sampled approximation $\hat{\mathbf{R}}^{k-1}$, which may deviate from the ground truth. This mismatch introduces a distribution shift that can propagate errors across stages in our multi-stage framework. To mitigate this, we propose conditional augmentation, which perturbs the ground truth with controlled noise during training, improving the generalization of the guidance mechanism and enhancing robustness at inference time.

For the training of stage $k$, we denote the coarse structure obtained from ground-truth data as $\mathbf{R}_0^{k-1}$. We perturb $\mathbf{R}_0^{k-1}$ to simulate the variability of coarse conformers encountered during inference. To be specific, we select a small timestep $s$ and perturb $\mathbf{R}_0^{k-1}$ leveraging the noise schedule of the diffusion forward process:

$$\mathbf{R}_s = \sqrt{\bar{\alpha}_s^{k-1}}\mathbf{R}_0^{k-1} + \sqrt{1 - \bar{\alpha}_s^{k-1}}\varepsilon, \quad \varepsilon \sim \mathcal{N}(0, \mathbf{I}), \tag{2}$$

where $\bar{\alpha}_s^{k-1}$ represents the cumulative noise schedule for stage $k-1$ at timestep $s$. This formulation indicates that perturbed structure $\mathbf{R}_s$ is sampled from $q(\mathbf{R}_s^{k-1}|\mathbf{R}_0^{k-1})$. Then the noisy coarse structure is passed through the molecular upsampling module $\mathcal{F}_k$ to generate a structurally aligned conditional input for the current stage:

$$\mathbf{R}_{\text{cond}}^k = \mathcal{F}_k(\mathbf{R}_s, \mathcal{G}^{k-1}, \mathcal{G}^k). \tag{3}$$

This method ensures that the fine-scale model learns to condition on controlled variations of coarse-scale structures, thereby improving robustness to structural diversity during inference.

**Training Loss.** We derive an evidence lower bound (ELBO) for our hierarchical framework under conditional augmentation. For simplicity, we ignore the molecular upsampling, which does not affect the conclusion. The following proposition provides an ELBO:

**Proposition 1.** *Assuming that the coarse-scale is unconditional and both the coarse and fine scale generative models use the same number of timesteps $T$, the ELBO of the combined model $p_\theta^s$ is*

$$-\log p_\theta^s(\mathbf{R}_0) \leq \mathbb{E}_q[L_T(\mathbf{R}_0^l)] + \sum_{t>s} D_{\text{KL}}(q(\mathbf{R}_{t-1}^l|\mathbf{R}_t^l, \mathbf{R}_0^l)||p_\theta(\mathbf{R}_{t-1}^l|\mathbf{R}_t^l, \mathcal{G}^l)) + L_\theta(\mathbf{R}_0|\mathbf{R}_s^l), \tag{4}$$

*where $\mathbf{R}_0^l$ is the original coarse-scale output, $\mathbf{R}_0$ is the fine-scale output, and $L_\theta(\mathbf{R}_0|\mathbf{R}_s^l, \mathcal{G}) = \mathbb{E}_q[L_T(\mathbf{R}_0)] + \sum_{t>1} D_{\text{KL}}(q(\mathbf{R}_{t-1}|\mathbf{R}_t, \mathbf{R}_0)||p_\theta(\mathbf{R}_{t-1}|\mathbf{R}_t, \mathbf{R}_s^l, \mathcal{G})) + L_\theta(\mathbf{R}_0|\mathbf{R}_1, \mathbf{R}_s^l, \mathcal{G})$.*

*Proof.* Refer to Appendix F.1. □

The above proposition formalizes the training objective of our framework. Optimizing the ELBO formulation in Eq. 4 decouples the training across different stages in the multi-scale generative process. This decoupling allows us to apply the diffusion loss at each stage independently. After adding the molecular upsampling, our loss function for stage $k$ is:

$$\mathcal{L}(\theta_k) = \mathbb{E}_{t, \mathbf{R}_0^k, \varepsilon}[||\varepsilon - \varepsilon_{\theta_k}(\mathbf{R}_t^k, \mathcal{G}_k, \mathbf{R}_{\text{cond}}^k, t)||] \tag{5}$$

For stage $k = 1$, the loss reduces to the standard denoising loss without condition. In practice, we can search multiple possible timesteps $s$ for conditional augmentation to achieve the best sample quality. However, it would be computationally expensive to retrain the model separately for each choice. To enhance the training efficiency, we formulate a set $T_s$ and adopt the same method in Ho et al. (2022a), which avoids retraining models by amortizing the model over uniform random $s \in T_s$ at training. We detail the training and sampling procedures in Algorithm 3 and Algorithm 4.

## 5 EXPERIMENT

In this section, we first describe the experimental setup, followed by comprehensive experiments that demonstrate the effectiveness of our framework. We also perform ablation studies to analyze the efficiency of our framework and the contributions of molecular upsampling and conditional augmentation. The visualizations of the generated molecules are provided in Appendix I.

## 5.1 Experiment Setup

**Datasets.** We conduct our experiments on the GEOM dataset (Axelrod & Gomez-Bombarelli, 2022), which contains GEOM-QM9 and GEOM-Drugs. Each dataset comprises 40,000 molecules for training and 5,000 molecules for validation, each containing 5 conformers following data split of Xu et al. (2022). For the test set, we selected 200 molecules for each dataset, resulting in 22,409 and 14,324 conformers in QM9 and Drugs. The details of the dataset are in Appendix C.1.

**Evaluation.** We adopted the widely used Coverage (COV) and Average Minimum (AMR) metrics proposed by Ganea et al. (2021) to assess the geometric quality of generated conformers within a threshold $\delta$. These metrics are based on root-mean-square deviation (RMSD), which is a normalized Frobenius norm between two atomic coordinate matrices aligned using the Kabsch algorithm Kabsch (1976). We also evaluate chemical properties of generated conformers, including the average energy $\bar{E}$, the lowest energy $E_{min}$, HOMO-LUMO gap $\epsilon$, the average gap $\overline{\triangle\epsilon}$, the minimum gap $\triangle\epsilon_{min}$, the maximum gap $\triangle\epsilon_{max}$. These properties reflect the chemical validity of a conformer.

**Baselines.** We primarily evaluate our framework by integrating it with GeoDiff (Xu et al., 2022), used as the main backbone in our experiments. To assess generalizability, we also apply it to ConfGF (Shi et al., 2021), ET-Flow, and EBD (Hassan et al., 2024). Models integrated with our framework are denoted by appending "+MSGEN" (e.g., GeoDiff+MSGEN). For broader comparison, we include RDKit (Landrum et al., 2013), which is arguably the most popular open-source software for conformer generation, and several deep generative models for conformer generation, such as GraphDG (Simm & Hernández-Lobato, 2019), CGCF (Xu et al., 2021a), ConfVAE (Xu et al., 2021b), and GeoMol (Ganea et al., 2021). Baseline results are adopted from Xu et al. (2022) and Zhang et al. (2024). We highlight the best results in tables using boldface.

**Framework Setting.** Our main experiment adopts a 2-stage framework, where each molecule is divided into heavy-atom backbones and hydrogen atoms. This design reflects chemical fact: heavy atoms provide a structural scaffold, while hydrogens influence local stereochemistry and fine-tune conformers (Clabbers et al., 2022; Thompson & White, 2023). The backbone is generated first to serve as a global spatial anchor, guiding hydrogen placement in the second stage. Beyond this setting, we also conduct experiments with a 3-stage framework that incorporates additional domain priors. The design guidance for incorporating additional stages is in Appendix E.

## 5.2 Performance Evaluation

Table 2: Geometric evaluation on GEOM-Drugs using different baselines ($\delta = 1.25\mathring{A}$)

| Models | COV-R(%) ↑ | | MAT-R($\mathring{A}$) ↓ | | COV-P(%) ↑ | | MAT-P($\mathring{A}$) ↓ | |
| --- | --- | --- | --- | --- | --- | --- | --- | --- |
| | Mean | Median | Mean | Median | Mean | Median | Mean | Median |
| RDkit | 45.74 | 31.75 | 1.5376 | 1.4004 | 54.79 | 59.48 | 1.3341 | 1.1996 |
| GraphDG | 8.27 | 0.00 | 1.9722 | 1.9845 | 2.08 | 0.00 | 2.4340 | 2.4100 |
| CGCF | 53.96 | 57.06 | 1.2487 | 1.2247 | 21.68 | 13.72 | 1.8671 | 1.8066 |
| ConfVAE | 55.20 | 59.43 | 1.2380 | 1.1417 | 22.96 | 14.05 | 1.8287 | 1.8159 |
| GeoMol | 67.16 | 71.71 | 1.0875 | 1.0586 | - | - | - | - |
| ConfGF | 62.15 | 70.93 | 1.1629 | 1.1596 | 23.42 | 15.52 | 1.7219 | 1.6863 |
| GeoDiff | 87.86 | 97.00 | 0.8686 | 0.8545 | 60.17 | 62.21 | 1.1871 | 1.1412 |
| GeoDiff+MSGEN | **90.41** | **98.16** | **0.8424** | **0.8248** | **66.26** | **70.65** | **1.1217** | **1.0682** |

**Geometric Evaluation.** The geometric evaluations of conformer generations are summarized in Table 2 and Table 9 (left in Appendix G). As shown in the Table, GeoDiff+MSGEN significantly improves GeoDiff across all metrics and achieves the best performance across different baselines. Specifically, the consistent improvement in precision metrics indicates that our model generates more accurate conformers. These results demonstrate that integrating our framework significantly enhances its structural awareness, enabling it to capture key substructures and generate high-quality molecular conformers, especially for complex molecules like those in GEOM-Drugs.

**Chemical Evaluation.** This task estimates the molecular *ensemble properties* (Axelrod & Gomez-Bombarelli, 2022) over a set of generated conformers. This can provide a direct assessment of the quality of generated samples. Following (Shi et al., 2021), we generated 50 samples for each of the 30 molecules, constituting a subset of QM9. Then we used the chemical tool PSI4 (Smith et al., 2020) to get the desired properties. The mean absolute errors (MAE) between calculated properties and the ground truth are reported in Table 3. We can observe that incorporating our framework significantly decreases the MAE across all evaluated ensemble properties for GeoDiff. Furthermore,

it shows that our framework enables the model to generate more chemically plausible molecular conformers, as evidenced by the significantly lower energy and HOMO-LUMO gap errors compared to baselines.

Table 3: MAE of different physical properties between generated and ground truth ensemble properties in eV.

| Method | $\bar{E}$ | $E_{min}$ | $\overline{\triangle\epsilon}$ | $\triangle\epsilon_{min}$ | $\triangle\epsilon_{max}$ |
|---|---|---|---|---|---|
| RDkit | 0.9233 | 0.6585 | 0.3698 | 0.8021 | 0.2359 |
| GraphDG | 9.1027 | 0.8882 | 1.7973 | 4.1743 | 0.4776 |
| CGCF | 28.9661 | 2.8410 | 2.8356 | 10.6361 | 0.5954 |
| ConFVAE | 8.2080 | 0.6100 | 1.6080 | 3.9111 | 0.2429 |
| ConFGF | 2.7886 | 0.1765 | 0.4688 | 2.1843 | **0.1433** |
| GeoDiff | 0.2597 | 0.1551 | 0.3091 | 0.7033 | 0.1909 |
| GeoDiff+MSGEN | **0.1795** | **0.1019** | **0.2035** | **0.5898** | 0.1755 |

Table 4: Geometric evaluation on GEOM-QM9 for domain generalization. ($\delta = 0.5$Å)

| Model | Train Data | COV-R(%) ↑ | | MAT-R(Å) ↓ | |
|---|---|---|---|---|---|
| | | Mean | Median | Mean | Median |
| GraphDG | QM9 | 73.33 | 84.21 | 0.4245 | 0.3973 |
| CGCF | QM9 | 78.05 | 82.48 | 0.4219 | 0.3900 |
| ConfVAE | QM9 | 77.84 | 88.20 | 0.4154 | 0.3739 |
| GeoMol | QM9 | 71.26 | 72.00 | 0.3731 | 0.3731 |
| SubgDiff | Drugs | 83.50 | 88.70 | 0.3116 | 0.3075 |
| GeoDiff | Drugs | 74.94 | 79.15 | 0.3492 | 0.3392 |
| GeoDiff+MSGEN | Drugs | **83.73** | **89.32** | **0.3055** | **0.2859** |

**Domain Generalization.** We conduct domain generalization experiments to demonstrate the effectiveness of our framework, following the design in SubgDiff (Zhang et al., 2024). Specifically, models are trained on GEOM-Drugs and evaluated on GEOM-QM9 without any fine-tuning. To ensure the validity of this setup, we systematically checked the two datasets at the molecule level using RDKit canonical SMILES and confirmed that they are disjoint. We also compare the results against several baseline models without domain transfer. As shown in Table 4, our method, GeoDiff+MSGEN, consistently outperforms the GeoDiff baseline across all evaluation metrics, achieving the best overall results in the table. Additional results for models trained on GEOM-Drugs and evaluated on GEOM-QM9 are provided in Appendix G.2, showing similar conclusions.

**Multi-Stage Scalability.** To demonstrate the scalability of our framework, we extend it to a 3-stage generation process. We introduced a new stage based on the MurckoScaffolds (Bemis & Murcko, 1996), which extracts core ring systems and linkers while removing side chains and decorations from the heavy-atom graph. The extracted subgraph provides a stable and chemically relevant foundation that reflects key molecular geometry. We integrate the 3-stage design with ET-Flow and GeoDiff, and evaluate on the GEOM-Drugs dataset. The result in Table 5 demonstrates that the 3-stage design provides an improvement over the 2-stage case. These improvements suggest that incorporating more chemistry-informed structural decomposition can further enhance model performance. They also highlight the potential of our framework in conformer generation, particularly for biologically relevant molecules (Schuffenhauer et al., 2007).

Table 5: Geometric evaluation of applying MSGEN across different baselines on GEOM-Drugs

| Backbone | Variant | $\delta(\text{Å})$ | COV-R(%) ↑ | | MAT-R(Å) ↓ | | COV-P(%) ↑ | | MAT-P(Å) ↓ | |
|---|---|---|---|---|---|---|---|---|---|---|
| | | | Mean | Median | Mean | Median | Mean | Median | Mean | Median |
| GeoDiff | baseline | 1.25 | 87.86 | 97.00 | 0.8686 | 0.8545 | 60.17 | 62.21 | 1.1871 | 1.1412 |
| | +MSGEN (2-stage) | | 90.41 | 98.16 | 0.8424 | **0.8248** | 66.26 | 70.65 | 1.1217 | 1.0682 |
| | +MSGEN (3-stage) | | **91.05** | **98.33** | 0.8410 | 0.8265 | **66.87** | **71.40** | **1.1147** | **1.0579** |
| ET-Flow | baseline | 0.75 | 74.47 | 81.06 | 0.5514 | 0.5288 | 55.21 | 54.22 | 0.7855 | 0.7341 |
| | +MSGEN (2-stage) | | 80.50 | 88.71 | 0.4579 | 0.4394 | 64.67 | 68.34 | 0.7342 | 0.6783 |
| | +MSGEN (3-stage) | | **81.91** | **89.24** | **0.4363** | **0.4280** | **66.12** | **69.51** | **0.7159** | **0.6573** |
| Conf-GF | baseline | 1.25 | 62.15 | 70.93 | 1.1629 | 1.1596 | 23.42 | 15.52 | 1.7219 | 1.6863 |
| | +MSGEN(2-stage) | | **65.51** | **73.31** | **1.1415** | **1.1328** | **27.72** | **20.21** | **1.6986** | **1.6765** |
| EBD | baseline | 1.25 | **92.10** | 98.40 | 0.8292 | 0.8391 | 65.97 | 67.95 | 1.1300 | 1.1026 |
| | +MSGEN(2-stage) | | 91.92 | **98.48** | **0.8257** | **0.8198** | **68.10** | **72.25** | **1.1013** | **1.0680** |

**Framework Generalizability.** To validate the generalizability of our framework in enhancing structural awareness, we integrate our framework with different generative models, such as Score Matching (Song & Ermon, 2019) and Flow Matching (Lipman et al., 2022). For instance, ConfGF is based on Score Matching, while ET-Flow utilizes Flow Matching. In addition, we also evaluate our framework with Equivariant Blurring Diffusion (EBD) (Park & Shen, 2024), which generates atomic details from coarse fragments. Following the 2-stage design, we integrate our framework and evaluate on the GEOM-Drugs dataset. Specifically, we use the data split and preprocessing method in ET-Flow since it utilizes a larger GEOM-Drugs dataset. As shown in Table 5, our framework consistently improves the generation quality across various kinds of generative models, generating more diverse and accurate conformers. This improvement highlights the effectiveness of our multi-

scale framework in enhancing structural awareness through progressive guidance, regardless of the underlying generative model. Further details of the implementation in Appendix D.

## 5.3 ABLATION STUDY

**Efficiency and Practicality** We further investigate the efficiency of the multi-scale design. In particular, we compare the performance of vanilla GeoDiff and our 2-stage MSGEN+GeoDiff under the same total number of diffusion steps. *We set the same number of steps for both stages*. From the result in Figure 3 and Table 14, we observe that GeoDiff+MSGEN consistently outperforms the baseline in most metrics, while spending a shorter average generation time for each molecule. This highlights that our framework leads to more efficient utilization of the diffusion step, improving both effectiveness and practicality. Additional analysis is in Appendix G.7.

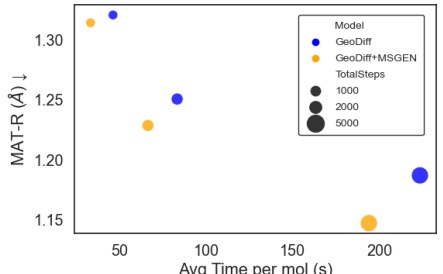

Figure 3: Comparison of GeoDiff and GeoDiff+MSGEN in terms of average generation time and mean MAT-R. Marker size corresponds to the total number of diffusion steps

**Impact of Hierarchical Design** To evaluate the effectiveness of the hierarchical design, we conduct a stage-wise ablation on GEOM-Drugs at the coarse level. Specifically, we compare the generation quality of coarse-level atoms generated (1) from scratch, (2) after Stage 1, and (3) after Stage 2.

| Stage | COV-R(%)↑ | MAT-R(Å)↓ | COV-P(%)↑ | MAT-P(Å)↓ |
|---|---|---|---|---|
| From scratch | 87.86 | 0.8686 | 60.17 | 1.1871 |
| After Stage 1 | 89.17 | 0.8543 | 65.32 | 1.1293 |
| After Stage 2 | 90.41 | 0.8424 | 66.26 | 1.1217 |

Table 6: Stage-wise geometric evaluation on coarse-level atoms of GEOM-Drugs.

As shown in Table 6, we observe consistent improvement from generation from scratch to Stage 1 and further to Stage 2. These results demonstrate that the hierarchical design enhances structural awareness by providing a meaningful coarse arrangement that can be further refined, rather than introducing rigid constraints.

**Impact of Molecular Upsampling.** To evaluate the effectiveness of molecular upsampling, we compare it with two alternatives: random sampling, which assigns hydrogen positions from a standard Gaussian, and centroid sampling, which places them at the geometric center of the coarse structure without noise. As shown in Figure 4a, our method yields more accurate conformers. This is primarily attributed to the effective utilization of chemical graph connectivity, which guides the fine-scale atomic placement based on anchor nodes, resulting in better structural guidance.

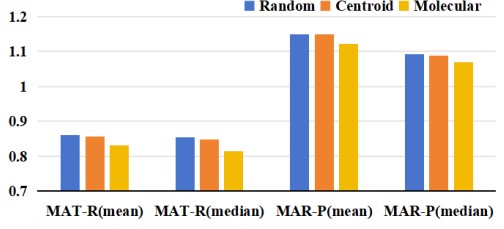
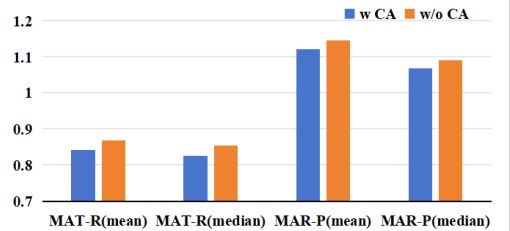

(a) Impact of Molecular Upsampling.  (b) Impact of Conditional Augmentation

Figure 4: Ablation studies on Molecular Upsampling and Conditional Augmentation.

**Impact of Conditional Augmentation.** To assess the performance of the conditional augmentation in our framework, we conduct an ablation study on GEOM-Drugs based on GeoDiff that removes the conditional augmentation in the training process. As shown in Figure 4b, conditional augmentation improves the performance of our framework, evidenced by lower MAT metrics. This demonstrates that this approach enhances the model's ability to adapt to varying structural scales and geometric variations, leading to more robust and accurate molecular generation. We further evaluate the

robustness and efficiency of MSGEN under reduced-step and equal-step settings. Additionally, we present a failure case analysis and a complexity-aware evaluation to better highlight the framework's strengths. We refer to Appendix G.3 to G.6 for detailed insights.

# 6 CONCLUSION

In this work, we introduce a novel multi-scale molecular conformer generation framework that leverages the inherent hierarchical structural organization of molecules to enhance structural awareness. Our framework generates coarse-grained substructures as intermediate representations, which guide finer-scale generation through a novel molecular upsampling strategy that bridges scales effectively. Through extensive experiments with competitive generative models, we substantiated the effectiveness and generalization ability of the proposed framework. Owing to its modular design, the framework flexibly incorporates domain-specific priors and is well-suited for complex molecular systems, including proteins and polymers, where hierarchical structures are intrinsic. These results suggest a promising direction for molecular conformer generation in increasingly complex chemical spaces.

# 7 ETHICS STATEMENT

This work develops machine learning methods for molecular conformer generation using publicly available datasets (GEOM-QM9 and GEOM-Drugs), which contain only computational molecular structures. No personal, sensitive data are involved, and no human or animal subjects are included.

The research aims to improve structural fidelity in conformer generation to support computational chemistry and drug discovery. While generative models may carry potential misuse risks, this study is limited to standard benchmarks of organic molecules and does not involve generating or evaluating bioactive or toxic compounds.

No real-world synthesis is performed, and no actionable synthesis protocols are provided. All results are intended solely for scientific benchmarking. We encourage continued attention to responsible use and safeguards against misuse in future work.

# 8 REPRODUCIBILITY STATEMENT

We have made extensive efforts to ensure the reproducibility of our work. All datasets used in our experiments are publicly available, and we provide a detailed description of preprocessing steps in Appendix C.1. The integration of our framework into baselines and the training hyperparameters are described in Appendix C.2, C.5, and D. All theoretical proofs are provided in the Appendix F. Code is available at: `https://github.com/Taserita/MSGEN-Full`.

# 9 ACKNOWLEDGEMENTS

Weizhi Gao, Zhichao Hou, and Xiaorui Liu are supported by the National Science Foundation (NSF) under grant number IIS-2443182. This work was completed in full during Jiapeng Hu's visit as a visiting scholar at North Carolina State University (NCSU).

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

## A  DENOISING DIFFUSION PROBABILISTIC MODEL

Denoising Diffusion Probabilistic Model(DDPM) (Ho et al., 2020) is defined by a forward process $q(x_t|x_{t-1})$ that gradually destroys data $x_0 \sim q(x_0)$ for $T$ timesteps and a learned reverse process $p_\theta(x_{t-1}|x_t)$ that iteratively denoises to recover the original data distribution. In the setting of molecular conformation generation, the diffusion model adds noise to molecular coordinates $\mathbf{R}$.

*The forward process* is defined as a fixed posterior distribution $q(\mathbf{R}_{1:T}|\mathbf{R}_0)$. Specifically, it is a Markov chain according to a fixed variance schedule $\beta_1, \beta_2, ..., \beta_T$:

$$q(\mathbf{R}_{1:T}|\mathbf{R}_0) = \prod_{t=1}^{T} q(\mathbf{R}_t|\mathbf{R}_{t-1}), \quad q(\mathbf{R}_t|\mathbf{R}_{t-1}) = \mathcal{N}(\mathbf{R}_t;\sqrt{1-\beta_t}\mathbf{R}_{t-1}, \beta_t\mathbf{I}) \tag{6}$$

Let $\alpha_t = 1 - \beta_t, \bar{\alpha}_t = \prod_{t=1}^{T}\beta_t$, so the closed form solution of the forward process $q(\mathbf{R}_t|\mathbf{R}_0)$ at arbitrary timestep $t$ is $q(\mathbf{R}_t|\mathbf{R}_0) = \mathcal{N}(\mathbf{R}_t;\sqrt{\bar{\alpha}_t}\mathbf{R}_{t-1}, (1-\bar{\alpha}_t)\mathbf{I})$. This indicates that with sufficiently large $T$, the whole forward process will convert $\mathbf{R}_0$ to a standard Gaussian distribution.

*The reverse process* is a reverse Markov chain of the above forward process starting from a Gaussian distribution $p(\mathbf{R}_T) = \mathcal{N}(\mathbf{R}_T;\mathbf{0},\mathbf{I})$:

$$p_\theta(\mathbf{R}_{0:T}|\mathcal{G}, \mathbf{R}_T) = \prod_{t=1}^{T} p_\theta(\mathbf{R}_{t-1}|\mathcal{G}, \mathbf{R}_t), \quad p_\theta(\mathbf{R}_{t-1}|\mathcal{G}, \mathbf{R}_t) = \mathcal{N}(\mathbf{R}_{t-1};\mu_\theta(\mathcal{G}, \mathbf{R}_t, t), \sigma_t^2\mathbf{I}) \tag{7}$$

Here $\mu_\theta$ are parameterized neural networks to estimate the means, and $\sigma_t$ can be any user-defined variance. In DDPM, $\mu_\theta = \frac{1}{\bar{\alpha}_t}(\mathbf{R_t} - \frac{\beta_t}{\sqrt{1-\bar{\alpha}_t}}\varepsilon_\theta(\mathcal{G}, \mathbf{R}_t, t))$, and $\varepsilon_\theta$ is the parameterized denoising network. Under the parameterization, we have the training objective of DDPM:

$$\mathcal{L}(\theta) = \mathbb{E}_{t,\mathbf{R_0},\varepsilon}[\|\varepsilon - \varepsilon_\theta(\mathcal{G}, \sqrt{\bar{\alpha}_t}\mathbf{R}_0 + \sqrt{1-\bar{\alpha}_t}\varepsilon, t)\|^2] \tag{8}$$

## B    EVALUATION METRICS

**Geometric Evaluation** We adopted the widely used metrics proposed by (Ganea et al., 2021) to assess the geometric quality of generated conformers. The metrics are based on root-mean-square deviation (RMSD), which is a normalized Frobenius norm between two atomic coordinate matrices aligned using the Kabsch algorithm (Kabsch, 1976). Given the reference conformer set $\mathcal{S}_r$ and the generated sample set $\mathcal{S}_g$, then the Coverage and Matching metrics following the conventional *Recall* measurement can be defined as:

$$\text{COV-R}(\mathcal{S}_g, \mathcal{S}_r) = \frac{1}{|\mathcal{S}_r|} \left| \left\{ \mathcal{C} \in \mathcal{S}_r | \text{RMSD}(\mathcal{C}, \hat{\mathcal{C}}) \leq \delta, \hat{\mathcal{C}} \in \mathcal{S}_g \right\} \right|, \tag{9}$$

$$\text{MAT-R}(\mathcal{S}_g, \mathcal{S}_r) = \frac{1}{|\mathcal{S}_r|} \sum_{\mathcal{C} \in \mathcal{S}_r} \min_{\hat{\mathcal{C}} \in \mathcal{S}_g} \text{RMSD}(\mathcal{C}, \hat{\mathcal{C}}), \tag{10}$$

where $\delta$ is the pre-defined threshold. The other two metrics COV-P and MAT-P is the *Precision* metric which can be defined similarly but with the generated and reference sets exchanged. In practice, $\mathcal{S}_g$ is set as twice of the size of $\mathcal{S}_r$ for each molecule. COV measures how well one set is covered by another within a threshold $\delta$, MAT measures the geometric closeness between two sets using the average RMSD of one conformer set with its closest conformer in another set. The recall metric is focused on the diversity, while the precision metric measures the quality.

**Chemical Evaluation** we also assess chemical properties of the generated conformers, including:

- Average energy $\bar{E}$: Measures the mean stability of generated conformers, calculated as the average of the total potential energy values of all generated conformers.
- Lowest energy $E_{min}$: Represents the minimum potential energy among all generated conformers, indicating the most stable structure.
- HOMO-LUMO gap $\epsilon$: The difference between the highest occupied molecular orbital (HOMO) and the lowest unoccupied molecular orbital (LUMO), reflecting electronic stability.
- The average gap $\overline{\triangle\epsilon}$: The mean value of the HOMO-LUMO gap across all conformers.
- The minimum gap $\triangle\epsilon_{min}$: The smallest gap among generated conformers, indicating the most chemically reactive structure.
- The maximum gap $\triangle\epsilon_{max}$: The largest gap, indicating the most electronically stable structure.

These chemical properties provide insights into the stability and chemical validity of the generated conformers, complementing the geometric quality assessments.

## C    IMPLEMENTATION DETAILS

### C.1    DATASETS

We used GEOM-QM9 (QM9) (Ramakrishnan et al., 2014) and GEOM-Drugs (Drugs) (Axelrod & Gomez-Bombarelli, 2022) for analysis and comparison between molecular conformer generation models. Each dataset comprises 40,000 molecules for the training set and 5,000 molecules for the validation set. We obtained the raw data, the pre-processed data and the data split at https://github.com/MinkaiXu/GeoDiff. Each dataset comprises 40,000 molecules for the training set and 5,000 molecules for the validation set, with each molecule containing 5 conformers. For the test set, we selected 200 molecules for each dataset, resulting in 22,409 and 14,324 conformers in QM9 and Drugs. Our primary focus is on GEOM-Drugs, the most extensive and chemically relevant subset comprising many drug-like molecules.

Table 7 summarizes the average number of nodes and edges per molecular graph in the datasets, both with and without hydrogen atoms. These statistics provide insight into the structural complexity and size differences between the two datasets.

### C.2    MODEL ARCHITECTURE

For the main experiment in the paper, we adopt the graph field network(GFN) from (Xu et al., 2022). Original GFN takes node embedding $h^l \in \mathbb{R}^{n \times b}$ ($b$ is the feature dimension) and corresponding

Table 7: Dataset Analysis

|  | GEOM-QM9 | GEOM-Drugs |
|---|---|---|
| Avg-node | 19.26 | 45.54 |
| Avg-node w/o H | 8.84 | 25.38 |
| Avg-edge | 38.75 | 95.13 |
| Avg-edge w/o H | 17.91 | 54.81 |

coordinate $x^l \in \mathbb{R}^{n \times 3}$ as input. It then produces $\mathbf{h}^{l+1}$, $\mathbf{x}^{l+1}$ as follows:

$$\mathbf{m}_{ij} = \Phi_m(\mathbf{h}_i^l, \mathbf{h}_j^l, ||\mathbf{x}_i^l - \mathbf{x}_j^l||^2, e_{ij}; \theta_m) \tag{11}$$

$$\mathbf{h}_i^{l+1} = \Phi_h(\mathbf{h}_i^l, \sum_{j \in \mathcal{N}(i)} \mathbf{m}_{ij}; \theta_h) \tag{12}$$

$$\mathbf{x}_i^{l+1} = \sum_{j \in \mathcal{N}(i)} \frac{1}{d_{ij}} (\mathbf{R}_i - \mathbf{R}_j) \Phi_x(\mathbf{m}_{ij}; \theta_x) \tag{13}$$

where $\Phi$ are feed-forward network and $d_{ij}$ denotes interatomic distances. $\mathcal{N}(i)$ denotes the neighborhood of $i_{th}$ node, including both connected atoms and other ones within radius threshold $t$, which enables the model to explicitly capture long-range interactions and support disconnected molecular graphs. To better adapt the network to our hierarchical generation setting, we only maintain the original GFN formulation in the first stage unchanged and modify the architecture in the later stages. The main difference is that we have extra information $x_p \in \mathbb{R}^{n \times 3}$ from the previous stage as input, so we change the formulation of $\mathbf{m}_{ij}$:

$$\mathbf{m}_{ij} = \Phi_m(h_i^l, h_j^l, ||\mathbf{x}_i^l - \mathbf{x}_j^l||^2, ||x_{p_i}^l - x_{p_j}^l||^2, e_{ij}; \theta_m) \tag{14}$$

## C.3  EQUIVARIANCE

Equivariance with respect to 3D transformations is a crucial property in molecular conformation generation, as the predicted conformations should be consistent under global rotations and translations of the input. The original GFN is translationally invariant and rotationally equivariant. Similarly, we can prove that the modified GFN also has the same properties.

**Proposition 2.** *Parameterizing our network $\varepsilon_\theta$ as a composition of L modified GFN layers, and take the $\mathbf{x}^L$ after L updates as the output. Then the network $\varepsilon_\theta$ is SE(3) equivariant w.r.t the 3D system $\mathbf{R}$.*

*Proof.* Refer to Appendix F.2. □

From Proposition 2, we construct an equivariant Markov kernel. If we then sample from the invariant distribution, we can obtain an equivariant reverse generative process using Proposition 3. We can show that at each stage, the probability distribution of the output is invariant and the output is equivariant.

**Proposition 3.** *(Xu et al., 2022) Let $p(x_T)$ be an SE(3)-invariant density function, $T_g$ be some roto-translational transformations from SE(3). If markov transition $p(x_{t-1}|x_t)$ are SE(3)-equivariant, i.e $p(x_{t-1}|x_t) = p(T_g(x_{t-1})|T_g(x_t))$, then we have the density $p_\theta(x_0)$ is also SE(3)-invariant.*

More importantly, we can prove that if each stage of the MSGEN framework is SE(3)-equivariant, then the output of the entire framework is also equivariant. This property is critical for ensuring that the generated molecular conformations remain physically meaningful under arbitrary global rotations and translations of the input.

**Proposition 4.** *Assume a MSGEN framework consist of N sequential neural networks $f_1, ..., f_N$, where each $f_i$ is equivariant under SE(3) transformations, then the full model $F = f_N \circ f_{N-1} \circ ... \circ f_1$ is also SE(3) equivariant.*

*Proof.* Refer to Appendix F.3. □

### C.4 Details of Preliminary Study

In this section, we provide the detailed setup and rationale for the preliminary study to evaluate the effectiveness of explicit geometric structural guidance in molecular generation. The primary goal of this study is to investigate whether incorporating explicit substructure information can significantly improve molecular conformer generation compared to existing methods.

**Model Selection**    To ensure a fair and comprehensive evaluation, we selected three representative generative models:

1. **GeoDiff**: A standard diffusion-based model that generates molecular conformers without explicitly leveraging structural features. It follows a conventional denoising diffusion process where atomic coordinates are progressively refined through Gaussian noise perturbation and denoising steps.

2. **SubgDiff**: An extension of the GeoDiff approach that incorporates a subgraph predictor for subgraph denoising, enhancing the model's ability to preserve subgraph patterns.

3. **EBD**: An Equivariant blurring diffusion model which generates atomic details from coarse-grained estimation of fragment structures, motivated by the blurring corruption of heat equation.

Additionally, we introduce **our variant model**, which builds on GeoDiff by explicitly incorporating structural guidance. Specifically, our model leverages *heavy-atom backbone positional information* as guidance during the diffusion process. We use the following strategies: 1) the heavy-atom backbone positions are directly obtained from the ground-truth data; 2) the remaining atoms we placed at the geometric center of the heavy atoms. This geometric center is calculated as the centroid of the heavy-atom positions and is considered as a *masked position* to indicate the absence of accurate atomic coordinates for these atoms.

**Experiment Setting**    To ensure a fair comparison between the selected models, we maintain consistent settings across all experiments. Specifically, we set the number of diffusion steps to 500 for all models. For GeoDiff (Xu et al., 2022) and our variant, we train both of them using the same hyperparameters in (Zhang et al., 2024). For SubgDiff (Zhang et al., 2024), we use the result from their paper. For EBD (Park & Shen, 2024), we train their model with 500 diffusion steps while keeping all other hyperparameters unchanged from the original implementation.

### C.5 Training and time

We used a single NVIDIA A6000 GPU for every training and generation task. For training, we used a learning rate $10^{-3}$ for the first stage and $10^{-4}$ for the second stage with the Adam (Diederik, 2014) optimizer. For GeoDiff, the practical training time of the first stage is around 24 hours, and the practical training time of the second stage is around 50 hours. The conditional augmentation time $s$ is set as $[1, 10]$, and other hyperparameters of our framework are summarized in Table 8, including highest variance level $\beta_t$, lowest variance level $\beta_1$, the variance schedule, number of diffusion timesteps $T$, radius threshold for determining the neighbor of atoms $\tau$, batch size, and number of training iterations.

Table 8: Additional hyperparameters of our framework

| Task | $\beta_1$ | $\beta_T$ | $\beta$ Scheduler | $T$ | $\tau(\mathring{A})$ | Batch Size | Training Iter. |
|------|-----------|-----------|-------------------|-----|----------------------|------------|----------------|
| QM9   | 1.e-7 | 2.e-3 | sigmoid | 5000 | 10.0 | 64 | 1.5M |
| Drugs | 1.e-7 | 2.e-3 | sigmoid | 5000 | 10.0 | 32 | 2M |
| QM9   | 1.e-7 | 2.e-2 | sigmoid | 500  | 10.0 | 64 | 1.5M |
| Drugs | 1.e-7 | 2.e-2 | sigmoid | 500  | 10.0 | 32 | 2M |
| Drugs | 1.e-7 | 9.e-3 | sigmoid | 1000 | 10.0 | 32 | 2M |
| Drugs | 1.e-7 | 5.e-3 | sigmoid | 2000 | 10.0 | 32 | 2M |
| Drugs | 1.e-7 | 4.e-3 | sigmoid | 2500 | 10.0 | 32 | 2M |

## C.6 TRAINING AND SAMPLING ALGORITHM

The following algorithms 3, 4 go over the pseudo-code for the two-stage training and sampling procedure. Following prior work (Xu et al., 2022), we sample from the zero center-of-mass system to build the invariant to rotation and translation sample density for each stage.

---

**Algorithm 3:** Training of a two-stage MSGEN framework

---

```
// Train base model
```
**repeat**
> $(\mathbf{R}^1, \mathcal{G}^1(\mathcal{V}, \mathcal{E}, \mathbf{X})) \sim p_1(\mathbf{R}^1, \mathcal{G}^1)$   `// Sample backbone of each molecule`
> $t \sim \mathcal{U}(\{1, .., T\})$
> $\varepsilon \sim \mathcal{N}(\mathbf{0}, \mathbf{I})$
> $\mathbf{R}_t^1 = \sqrt{\bar{\alpha}_t} \mathbf{R}^1 + \sqrt{1 - \bar{\alpha}_t} \varepsilon$
> $\theta \leftarrow \theta - \eta \nabla_\theta \left\| \varepsilon - \varepsilon_\theta(\mathbf{R}_t^1, \mathcal{G}^1, t) \right\|^2$

**until** converged
```
// Train second-stage model(in parallel with base model)
```
**repeat**
> $(\mathbf{R}, \mathbf{R}^1, \mathcal{G}(\mathcal{V}, \mathcal{E}, \mathbf{X})) \sim p(\mathbf{R}, \mathcal{G})$   `// Sample molecule and get the`
>   `backbone`
> $s \sim \mathcal{U}(\{1, .., T_s\}), t \sim \mathcal{U}(\{1, .., T\})$
> $\varepsilon, \varepsilon_1 \sim \mathcal{N}(\mathbf{0}, \mathbf{I})$
> $\mathbf{R}_s^1 = \sqrt{\bar{\alpha}_s} \mathbf{R}^1 + \sqrt{1 - \bar{\alpha}_s} \varepsilon_1$
> $\mathbf{R}_t = \sqrt{\bar{\alpha}_t} \mathbf{R} + \sqrt{1 - \bar{\alpha}_t} \varepsilon$
> $\mathbf{R}_{\text{cond}}^2 \leftarrow \text{Upsampling}(\mathbf{R}_s^1, \mathcal{G}^1, \mathcal{G})$   `// Generate guidance`
> $\theta \leftarrow \theta - \eta \nabla_\theta \left\| \varepsilon - \varepsilon_\theta(\mathbf{R}_t, \mathbf{R}_{\text{cond}}^2, \mathcal{G}, t) \right\|^2$

**until** converged

---

**Algorithm 4:** Sampling of a two-stage MSGEN framework

---

**Input:** the molecular graph $\mathcal{G}(\mathcal{V}, \mathcal{E}, \mathbf{X}), \mathcal{G}^1(\mathcal{V}, \mathcal{E}, \mathbf{X})$, the diffusion step $T_1, T_2$
**Output:** the 3D molecular conformation $\mathbf{R}_0$
$\mathbf{R}_{T_1}^1 \sim \mathcal{N}(\mathbf{0}_1, \mathbf{I}_1)$
**for** $i \leftarrow T_1$ **to** 1 **do**
> Shift $\mathbf{R}_i^1$ to zero CoM
> $\mathbf{R}_{i-1}^1 \sim p_\theta(\mathbf{R}_{i-1}^1 | \mathbf{R}_i^1, \mathcal{G}^1)$

**end**
$\mathbf{R}_{\text{cond}}^2 \leftarrow \text{Upsampling}(\mathbf{R}_0^1, \mathcal{G}^1, \mathcal{G})$   `// Generate structural prior`
$\mathbf{R}_{T_2} \sim \mathcal{N}(\mathbf{0}, \mathbf{I})$
**for** $i \leftarrow T_2$ **to** 1 **do**
> Shift $\mathbf{R}_i$ to zero CoM
> $\mathbf{R}_{i-1} \sim p_\theta(\mathbf{R}_{i-1} | \mathbf{R}_i, \mathbf{R}_{\text{cond}}^2, \mathcal{G})$

**end**
**return** $\mathbf{R}_0$

---

## C.7 UPSAMPLING MODULES

In our framework, we design several modules for molecular structure refinement across scales. Below, we present the implementation details of both techniques.

### C.7.1 CENTROID AND RANDOM UPSAMPLING

**Centroid Upsampling.** The additional fine-level atoms are placed at the geometric center of the entire coarse structure. This geometric center is calculated as the centroid of the coarse structural positions. It is considered as a *masked position* to indicate the absence of accurate atomic coordinates for these atoms.

**Random Upsampling** The spatial coordinates of the additional fine-level atoms are randomly drawn from a standard Gaussian distribution. It is a random initialization for unknown atoms.

### C.7.2 MOLECULAR UPSAMPLING

This method(See Algorithm 1, 2) reconstructs fine-level atomic coordinates by leveraging coarse-grained atoms as chemically meaningful anchors. For each fine-grained atom—such as a hydrogen or side-chain atom—its spatial position is estimated relative to its covalently bonded atoms. Random directions are applied within a chemically plausible radius to reflect local variations, ensuring the reconstructed geometries respect chemical bonding patterns and three-dimensional spatial constraints.

### C.7.3 GEOMETRIC UPSAMPLING

Our primary framework is designed to leverage explicit substructural guidance as it provides superior accuracy in generating molecular conformers, as demonstrated by our study. However, we recognize that there are practical scenarios where coarse features are used. In such cases, it is beneficial to have a flexible alternative that can still provide reasonable geometric reconstructions.

To address this, we propose a method(See Algorithm 5) that reconstructs fine-level atomic coordinates by interpreting each coarse-level site as the centroid of a chemically meaningful structural region, such as a rigid substructure center or functional group. We define a local conformational region $\mathcal{S}(i, \tau)$ around each coarse node $i$, modeled as a sphere in 3D space, within which the fine-level atoms are distributed. To capture the spatial variability, atom positions are sampled along random directions on the surface of this sphere, with distances scaled by a chemically plausible radius $\tau$.

---

**Algorithm 5:** Structure-Aware Geometric Upsampling

---

**Input:** Coarse coordinates $\mathbf{R}_{\text{coarse}} \in \mathbb{R}^{n \times 3}$, mapping $\mathcal{M}$ from fine atom $j$ to coarse index $i$, radius prior $\tau$
**Output:** Fine-grained conditions $\mathbf{R}_{\text{cond}} \in \mathbb{R}^{m \times 3}$
Initialize $\mathbf{R}_{\text{cond}} \leftarrow \mathbf{0}$;
**for** *Each fine-level atom $j = 1, \ldots, m$* **do**
    Get parent coarse node $i \leftarrow \mathcal{M}(j)$;
    Sample direction $\mathbf{d}_j \sim \mathcal{N}(\mathbf{0}, \mathbf{I})$;
    Normalize $\mathbf{d}_j \leftarrow \mathbf{d}_j / \|\mathbf{d}_j\|$;
    $\mathbf{R}_{\text{cond}}[j] \leftarrow \mathbf{R}_{\text{coarse}}[i] + r_j \cdot \mathbf{d}_j$
**end**
**return** $\mathbf{R}_{\text{cond}}$

---

In practice, given the coarse feature positions $\mathbf{R}_{\text{coarse}} \in \mathbb{R}^{n \times 3}$ and the coarse-to-fine mapping $\mathcal{M}$, our objective is to reconstruct the atom-level conditions $\mathbf{R}_{\text{cond}} \in \mathbb{R}^{m \times 3}$. For each fine-level atom $j$, we use the mapping $\mathcal{M}$ to find its related coarse node $i = \mathcal{M}(j)$. And then sample a random direction $d_j \sim \mathcal{N}(0, \mathbf{I})$ and normalize it. The final conditional position of atom $j$ is place on the tangent space of conformational sphere $\mathcal{S}(i, \tau)$ by:

$$\mathbf{R}_{\text{cond}}[j] = \mathbf{R}_{\text{coarse}}[i] + \tau \cdot d_j, \tag{15}$$

This method reflects the spatial flexibility of atoms anchored around coarse node, and enables geometry reconstruction even without explicit bonding constraints. This idea is inspired by prior efforts in (Riniker & Landrum, 2015; Qiang et al., 2023), where spatial priors are incorporated to generate more realistic molecular conformers. We prove the effectiveness of this method in Appendix G.9

**Coarse-to-Coarse Conditioning via Atom-Level Reconstruction.** While our primary approach emphasizes explicit substructure guidance, there may be scenarios where both the current and previous stages operate at coarse resolutions (e.g., fragments or virtual nodes). When both the current and previous stages operate at coarse resolutions, we adopt a reconstruction-based strategy to ensure meaningful cross-stage conditioning. Specifically, the coarse output $\mathbf{R}^{(k-1)}$ of stage $k - 1$ is first upsampled to recover an approximate full-atom structure using chemical and geometric upsampling. This full-resolution conformation is then re-coarsened using the same rules $\mathcal{M}_k$ as stage $k$, producing a resolution-aligned structure that serves as

$$\mathbf{R}_{\text{cond}}^{(k)} = \mathcal{M}_k(\mathcal{F}(\mathbf{R}^{(k-1)}, \mathcal{G}^{(k-1)}, \mathcal{G}^{(k)})).$$

This strategy preserves chemical plausibility while ensuring geometric consistency across coarse levels, acting as a structural regularizer that improves alignment and stability throughout the generative framework.

**Unified perspective** The rationale behind introducing the geometric upsampling and coarse-to-coarse conditioning strategies is to enhance the flexibility and extensibility of our framework. While our primary framework is designed to leverage explicit substructural guidance, we recognize that practical scenarios often demand flexibility in handling diverse input types. In particular, there are situations where coarse features are the only available structural cues, or where transitions between coarse generative stages are necessary. The introduction of these methods highlights our framework's compatibility with both explicit and coarse features, showing its broad applicability.

## D   IMPLEMENTATION OF FRAMEWORK GENERALIZATION

**Dataset.** For ConfGF (Shi et al., 2021), we use the same dataset as GeoDiff. And for ET-Flow (Hassan et al., 2024), we use the GEOM-Drugs dataset from (Axelrod & Gomez-Bombarelli, 2022), which contains 304k drug-like molecules, each with an average of 44 atoms. For the data split, we use a train/validation/test (243473/30433/1000) split following the setting in their paper.

**Baseline.** For ConfGF, we use the result from their paper. And for ET-Flow, we download their code from the public Github https://github.com/shenoynikhil/ETFlow and use their config to train their model for our experiment. Notably, during the reproduction of ET-Flow (Hassan et al., 2024), we encountered a critical issue when integrating the original source code. Specifically, the codebase utilizes the Kabsch algorithm to perform alignment during training and samples from a harmonic prior. However, we found that this setup frequently results in NaN values during the initial training iterations, which renders the model training process. It is worth noting that the ET-Flow itself supports sampling from a Gaussian distribution as an alternative to the harmonic prior. Therefore, to ensure stable training, we used the Gaussian distribution throughout our experiments.

To maintain the consistency and fairness of our evaluation, we preserved all other experimental settings and hyperparameters as described in the original ET-Flow implementation. The only adjustments were the use of Gaussian sampling and the removal of the alignment step. As a result, the reported performance of ET-Flow in our work may differ from the originally reported results. However, these results represent a reasonable and reproducible implementation of the ET-Flow.

### D.1   MODEL ARCHITECTURE

**ET-Flow**   Original ET-Flow consists of two main components:(1) a representation module based on the equivariant transformer architecture from TorchMD-NET (Thölke & De Fabritiis, 2022) and (2) the equivariant vector output module (Schütt et al., 2017). We prefer readers to (Hassan et al., 2024) for more details.

For the conditional case, we modify the neighborhood embedding layer, which captures the local atomic environment:

$$n_i = \sum_{j=1}^{N} embed^{nbh}(z_j) \cdot [g(d_{ij}, l_{ij}), g(d_{ij}^c, l_{ij})]. \tag{16}$$

where $embed^{nbh}(z_j)$ provides a separate embedding for atomic numbers, $d_{ij}$ is the distance between atom $i$ and $j$, $d_{ij}^c$ is the distance from the condition structure and $l_{ij}$ encodes edge features. The interaction function $g$ combines distance and edge information. Then we change the attention weight formula:

$$A = SiLU(dot(Q, K, D^K)) \cdot \phi(d_{ij}, d_{ij}^c) \tag{17}$$

**ConfGF**   ConfGF adopts a variant of Graph Isomorphism Network (GINs) (Xu et al., 2018) to compute node embedding based on graph structures. And then use an MLP as the noise conditional network to map edge embedding to scores. We prefer readers to (Shi et al., 2021) for more details.

For the conditional case, given molecular graph $\mathcal{G} = (\mathcal{V}, \mathcal{E})$, wo modify the edge embedding layer:

$$h_{e_{ij}} = \text{MLP}(e_{ij}, d_{ij}, d_{ij}^{cond}), \forall e_{ij} \in \mathcal{E} \tag{18}$$

Here $e_{ij}$ is the edge attribute and $d_{ij}^{cond}$ is calculated from the condition.

### D.2 Equivariant Blurring Diffusion

Equivariant Blurring Diffusion (EBD) (Park & Shen, 2024) has demonstrated promising performance in generating molecular conformers by progressively refining coarse molecular features into fine-grained atomic details. However, its reliance on fixed coarse fragments as starting points limits its structural adaptivity, especially when generating large and flexible molecules where maintaining spatial coherence among substructures is crucial. Our proposed multi-scale molecular generation framework provides an opportunity to address this limitation and enhance the performance of EBD.

**Analysis of EBD.** EBD leverages a blurring schedule as the core mechanism for generating molecular conformers. To understand why EBD can be seamlessly integrated with our multi-scale framework, it is essential to first analyze the fundamental principle behind its blurring process:

$$f_{\mathbf{B}}(x_0^a, \hat{x}^f, t) = (1 - \frac{t}{T})x_0^a + \frac{t}{T}\mathbf{M}\hat{x}^f \tag{19}$$

where $x_0^a$ is the ground-truth atom position, $\hat{x}^f$ is the fragment feature, and $\mathbf{M}$ is the projection matrix between fragments and atoms. So $\mathbf{M}\hat{x}^f$ represents the positional prior derived from fragment features, serving as the initialization for the relative atomic positions within each fragment. In other words, EBD essentially propagates the atomic positions outward from the fragment centers, progressively reducing ambiguity to obtain the final conformer. Our framework can provide a better structural prior for this process.

**Experiment Setting for EBD.** For the baseline model, we use the checkpoint from the public Github https://github.com/Shen-Lab/EBD. To enhance structural awareness of EBD, we design a two-stage framework that integrates EBD with our method. In the first stage, we leverage GeoDiff (Xu et al., 2022) to generate the heavy-atom backbone, which serves as a structural scaffold. We then apply our chemical upsampling method to build a more informative condition for EBD progressively. Specifically, we utilize the backmapping matrix $\mathbf{M}^{\dagger}$ to transform the upsampled condition into a coarse-level feature $\hat{x}^f$. In the second stage, we employ EBD to refine the atomic coordinates from the coarse features obtained in the first stage. To ensure that EBD effectively leverages the new coarse features, we retrain EBD with the same hyperparameters from scratch using these enhanced initial conditions. This retraining process adapts EBD to the improved structural priors, enabling it to recover fine-grained atomic details more accurately.

**Discussion** Since the accuracy of the initial atomic positions is critical to the final conformer quality, the fragment-based initialization in EBD can limit its performance, especially when generating large and flexible molecules. Our framework addresses this limitation by providing more accurate and chemically plausible priors through a multi-stage generation process. Our framework can provide atomic coordinates that are better aligned with the underlying molecular structure, which shows the superiority.

## E Detailed Guidance on Multi-Scale Design

We provide guidance on how MSGEN can be extended to multi-scale design. The MSGEN framework is designed to seamlessly integrate with existing molecular decomposition strategies and adapt to diverse application needs. In particular, additional coarse stages can be introduced to capture more fine-grained structural priors. We outline two general approaches:

- **Graph-based extraction.** Standard chemical decomposition methods such as MurckoScaffolds (Bemis & Murcko, 1996), BRICS (Degen et al., 2008), or other domain-specific motifs can be applied to extract chemically meaningful substructures (e.g., core ring systems, linkers). These substructures then serve as coarse representations for the first stage of generation.

- **Structure-based construction.** Coarse scaffolds can also be directly constructed from SMILES strings or RDKit (Landrum et al., 2013) `RDMol` objects, which is particularly useful when the scaffold is predefined by synthetic or functional constraints.

Our structure-aware molecular upsampling then bridges each stage by propagating geometry from coarse to fine levels. This modular design allows our framework to flexibly accommodate both data-driven and manually designed structural hierarchies, making it applicable across a wide range of molecular generation tasks.

## F PROOFS

### F.1 PROOF OF PROPOSITION 1

*Proof.* In the proposition, the ELBO is derived by conditioning directly on the coarse structure $\mathbf{R}_s^l$, which can be considered as truncating the reverse process of coarse scale at time $s$, without applying the upsampling module $\mathcal{F}_k$. This simplification preserves analytical tractability while retaining the essential coarse-to-fine dependency. Recall that the original evidence lower bound(ELBO) $-\log p_\theta(x_0) \leq L_\theta(x_0)$ in DDPM, where

$$L_\theta(x_0) = \mathbb{E}_q[L_T(x_0) + \sum_{t>1} D_{\mathrm{KL}}(q(x_{t-1}|x_t,x_0)||p_\theta(x_{t-1}|x_t)) - \log p_\theta(x_0|x_1)], x_0 \sim q(x_0) \quad (20)$$

So when we generate a finer-scale molecule $\mathbf{R}_0$ involving first generating the coarser-scale molecule $\mathbf{R}_0^l$ from the previous stage model $p_\theta(\mathbf{R}_0^l, \mathcal{G}^l)$ and then feeding that result into $p_\theta(\mathbf{R}_0|\mathbf{R}_0^l, \mathcal{G})$. So this can be considered as using ancestral sampling from the latent variable model:

$$p_\theta(\mathbf{R}_0) = \int p_\theta(\mathbf{R}_0|\mathbf{R}_0^l, \mathcal{G})p_\theta(\mathbf{R}_0^l, \mathcal{G}^l)d\mathbf{R}_0^l = \int p_\theta(\mathbf{R}_0|\mathbf{R}_0^l, \mathcal{G})p_\theta(\mathbf{R}_{0:T}^l, \mathcal{G}^l)d\mathbf{R}_{0:T}^l \quad (21)$$

So, using the conditional augmentation can be considered as truncating the previous generative process at timestep $s > 0$; i.e

$$p_\theta^s(\mathbf{R}_0) = \int p_\theta(\mathbf{R}_0|\mathbf{R}_s^l, \mathcal{G})p_\theta(\mathbf{R}_s^l, \mathcal{G}^l)d\mathbf{R}_s^l = \int p_\theta(\mathbf{R}_0|\mathbf{R}_s^l, \mathcal{G})p_\theta(\mathbf{R}_{s:T}^l, \mathcal{G}^l)d\mathbf{R}_{s:T}^l \quad (22)$$

This can be considered as a form of data augmentation since we condition on noisy data. Then let us examine the ELBO for $p_\theta^s(\mathbf{R}_0)$ in Eq. equation 22. Then we treat $p_\theta^s(\mathbf{R}_0)$ as a VAE (Kingma et al., 2013) with diffusion model prior, and the approximate posterior distribution:

$$q(\mathbf{R}_{0:T}, \mathbf{R}_{0:T}^l|\mathbf{R}_0, \mathbf{R}_0^l) = \prod_{t=1}^T q(\mathbf{R}_t|\mathbf{R}_{t-1})q(\mathbf{R}_t^l|\mathbf{R}_{t-1}^l) \quad (23)$$

which executes separate forward processes for the two-stage pairs. So we have the new ELBO using Eq. equation 20:

$$-\log p_\theta^s(\mathbf{R}_0) \leq \mathbb{E}_q[[L_T(\mathbf{R}_0^l) + \sum_{t>s} D_{\mathrm{KL}}(q(\mathbf{R}_{t-1}^l|\mathbf{R}_t^l, \mathbf{R}_0^l)||p_\theta(\mathbf{R}_{t-1}^l|\mathbf{R}_t^l, \mathcal{G}^l)) - \log p_\theta(\mathbf{R}_0|\mathbf{R}_s^l, \mathcal{G})]$$

$$(24)$$

where $L_T(\mathbf{R}_0) = D_{\mathrm{KL}}(q(\mathbf{R}_T|\mathbf{R}_0)||p(\mathbf{R}_T))$. Note that the last term $-\log p_\theta(\mathbf{R}_0|\mathbf{R}_s^l, \mathcal{G})$ can also consider as a latent variable model conditioned on $\mathbf{R}_s^l$. So it has a ELBO of the form $-\log p_\theta(\mathbf{R}_0|\mathbf{R}_s^l, \mathcal{G}) \leq L_\theta(\mathbf{R}_0|\mathbf{R}_s^l, \mathcal{G})$:

$$L_\theta(\mathbf{R}_0|\mathbf{R}_s^l, \mathcal{G}) = \mathbb{E}_q[L_T(\mathbf{R}_0) + \sum_{t>1} D_{\mathrm{KL}}(q(\mathbf{R}_{t-1}|\mathbf{R}_t, \mathbf{R}_0)||p_\theta(\mathbf{R}_{t-1}|\mathbf{R}_t, \mathbf{R}_s^l, \mathcal{G})$$
$$- \log p_\theta(\mathbf{R}_0|\mathbf{R}_1, \mathbf{R}_s^l, \mathcal{G})]. \quad (25)$$

Thus, taking Eq. equation 25 into Eq. equation 24, we have the ELBO for the combined model and finish our proof. $\square$

### F.2 PROOF OF PROPOSITION 2

*Proof.* We will prove that the output $\mathbf{x}$ of our network defined in equation 12, 13 and 14 is translationally invariant and rotationally equivariant with the input $\mathbf{R}$, which means

$$R\mathbf{x}^{l+1}, \mathbf{h}^{l+1} = \mathrm{GFN}(R\mathbf{x}^l, R\mathbf{R} + g, \mathbf{h}^l) \quad (26)$$

where $g \in \mathbb{R}^3$ denote any translation transformations and orthogonal matrices $R \in \mathbb{R}^{n \times 3}$ denote any rotation transformations.

Firstly, the node feature $\mathbf{h}^l$ is already invariant to SE(3) transformations, and the distances calculated between two atoms $||\mathbf{x}_i^l - \mathbf{x}_j^l||^2, ||x_{p_i}^l - x_{p_j}^l||^2$ are still invariant to rotations. Because

$$||R\mathbf{x}_i - R\mathbf{x}_j||^2 = (\mathbf{x}_i - \mathbf{x}_j)^T R^T R(\mathbf{x}_i - \mathbf{x}_j) = (\mathbf{x}_i - \mathbf{x}_j)^T I(\mathbf{x}_i - \mathbf{x}_j) = ||\mathbf{x}_i^l - \mathbf{x}_j^l||^2 \quad (27)$$

So the $\mathbf{m}_{ij}$ in equation 14 is invariant. Similarly, we can prove that $\mathbf{h}^{l+1}$ from equation 12 will also be invariant.

Next, we prove that the position vector $\mathbf{x}^{l+1}$ updated by equation 13 preserves rotation equivariance and translational invariance. Given $\mathbf{m}_{ij}$ already invariant as proven before, we have:

$$\sum_{j \in \mathcal{N}(i)} \frac{1}{d_{ij}}(R\mathbf{R}_i + g - R\mathbf{R}_j - g)\Phi_x(\mathbf{m}_{ij}) = R \sum_{j \in \mathcal{N}(i)} \frac{1}{d_{ij}}(\mathbf{R}_i - \mathbf{R}_j)\Phi_x(\mathbf{m}_{ij}) \quad (28)$$

Therefore, we show that $\mathbf{x}^{l+1}$ satisfies the properties and finish our proof. And by induction, it can be generalized to the L-layer case. □

### F.3 PROOF OF PROPOSITION 4

*Proof.* We can prove this by induction on the number of composed functions. Let $\mathbf{R}_0 = \mathbf{R}$ and define $\mathbf{R}_k := f_k(\mathbf{R}_{k-1})$, so that $F(\mathbf{R}_0) = \mathbf{R}_k$. Then we proceed by induction:

1. $n = 1$, the conclusion is clearly established by the assumption.

2. Assume that the statement holds for $n = k$, i.e.,

$$f_n \circ f_{n-1} \circ \cdots \circ f_1(R\mathbf{R}_0 + g) = R\mathbf{R}_n + g.$$

Then, using the equivariance of $f_{n+1}$ and the inductive hypothesis, we have:

$$\begin{aligned}
\mathbf{R}_{n+1}^{(R,g)} &= f_{n+1}(f_n \circ \cdots \circ f_1(R\mathbf{R}_0 + g)) \\
&= f_{n+1}(R\mathbf{R}_n + g) \\
&= Rf_{n+1}(\mathbf{R}_n) + g = R\mathbf{R}_{n+1} + g.
\end{aligned}$$

Therefore, the equivariance property holds for $k = n + 1$.

By induction, the full model $F_N = f_N \circ f_{N-1} \circ \cdots \circ f_1$ is SE(3) equivariant.

□

## G ADDITIONAL EXPERIMENT RESULTS

### G.1 CONFORMER GENERATION ON GEOM-QM9

We report the performance of GeoDiff+MSGEN compared to several baselines on the GEOM-QM9 dataset in Table 9. GeoDiff+MSGEN achieves the best or highly competitive results across almost all evaluation metrics.

**Results and Discussion**   Compared with the result in Table 2 on GEOM-Drugs, we can see the improvement margins are relatively smaller compared to results on the larger and more flexible GEOM-Drugs dataset. This is primarily because molecules in GEOM-QM9 are generally smaller, structurally simpler, and exhibit limited conformational flexibility. Due to the lower complexity of the underlying structures, even models with less sophisticated structural reasoning can achieve competitive results, thereby narrowing the observed performance gap.

In contrast, GEOM-Drugs contains larger and significantly more flexible molecules, with complex multi-scale conformational patterns that pose a much greater challenge. In this regime, our framework strength the perception of both global molecular topology and local structural variations and achieve much larger improvements over baselines.

Table 9: Geometric evaluation on GEOM-QM9 ($\delta = 0.5A$)

| Models | COV-R(%) ↑ Mean Median | MAT-R($\mathring{A}$) ↓ Mean Median | COV-P(%) ↑ Mean Median | MAT-P($\mathring{A}$) ↓ Mean Median |
|---|---|---|---|---|
| GraphDG | 73.33 84.21 | 0.4245 0.3973 | 43.90 35.33 | 0.5809 0.5823 |
| CGCF | 78.05 82.48 | 0.4219 0.3900 | 36.49 33.57 | 0.6615 0.6427 |
| ConfVAE | 55.84 88.20 | 0.4154 0.3739 | 38.02 34.67 | 0.6215 0.6091 |
| GeoMol | 71.26 72.00 | 0.3731 0.3731 | – – | – – |
| ConfGF | 88.49 94.31 | 0.2673 0.2685 | 46.43 43.41 | 0.5224 0.5124 |
| SubgDiff | **90.91** 95.59 | 0.2460 0.2351 | 50.16 48.01 | 0.6114 0.4791 |
| GeoDiff | 88.20 91.20 | 0.2435 **0.2211** | 51.16 50.53 | 0.5019 0.4413 |
| GeoDiff+MSGEN | 88.90 **95.73** | **0.2428** 0.2279 | **52.10 50.98** | **0.5014 0.4222** |

## G.2 DOMAIN GENERALIZATION ON GEOM-DRUGS

We further evaluate the domain generalization ability of our method by training on QM9(small molecular with up to 9 heavy atoms) and testing on Drugs(medium-sized organic compounds).

Table 10: Results on the GEOM-Drugs dataset. ($\delta = 1.25A$)

| Model | Train Data | COV-R(%) ↑ Mean | Median | MAT-R(Å) ↓ Mean | Median |
|---|---|---|---|---|---|
| GeoDiff | QM9 | 7.99 | 0.00 | 1.9722 | 1.9845 |
| GeoDiff + MSGEN | QM9 | **22.11** | **12.71** | **1.7924** | **1.7200** |

**Results and Discussion** The results can be found in Table 10. As shown in the results, our framework significantly improves the performance of the baseline model, demonstrating that our framework effectively enhances the robustness and generalization of the baseline model.

## G.3 FAILURE CASE ANALYSIS

While the MSGEN framework generally improves generation quality across datasets, we also examined whether there exist systematic failure cases where the framework could degrade conformer quality compared to the vanilla baseline GeoDiff.

**Results and Discussion** As summarized in Table 11, we identified a small number of molecules (9 out of 200) where incorporating MSGEN yielded higher MAT compared to the baseline. Visualization and chemistry-aware interpretation suggest these molecules often contain multiple ring systems connected by long and flexible linkers. Such configurations exhibit large conformational variability, and small deviations in positional guidance may lead to misaligned ring orientations or distorted global geometry. Additionally, several failure molecules already displayed high MAT under the baseline, indicating that these cases are inherently challenging to generate for GeoDiff. Thus, the observed degradation may not primarily stem from the framework, but rather reflect fundamental challenges in modeling highly flexible, multi-ring structures. Despite these rare cases, the majority of molecules benefit from the structural prior introduced by MSGEN, and the overall performance trends remain consistently favorable.

| $\Delta$MAT (Å) | Count | Interpretation |
|---|---|---|
| $< 0$ | 157 | Improvement |
| $0 \sim 0.1$ | 34 | Slight degradation |
| $> 0.1$ | 9 | Failure |
| **Total** | 200 | |

Table 11: Analysis of potential failure cases in generation.

## G.4 Few Step Generation

To evaluate the effectiveness of our multi-scale generation framework, we conduct an ablation study in which the whole diffusion process in each stage is replaced with a few-step generation scheme. Specifically, we reduce the number of denoising steps in each stage $T$ to a small fixed value and slightly adjust the noise schedule while keeping the overall architecture and training setup unchanged. For the above baselines, we reuse the results reported in (Zhang et al., 2024) for GeoDiff. For Subgdiff, we train their model based on the config released on the public GitHub and get the result.

| Dataset | Model | COV-R(%) ↑ | | MAT-R(Å) ↓ | | COV-P(%) ↑ | | MAT-P(Å) ↓ | |
|---------|-------|------|--------|------|--------|------|--------|------|--------|
| | | Mean | Median | Mean | Median | Mean | Median | Mean | Median |
| Drugs (T=1000) | GeoDiff | 82.96 | 96.29 | 0.9525 | 0.9334 | 48.21 | 46.03 | 1.3205 | 1.2824 |
| | GeoDiff+MSGEN | **90.05** | **98.00** | **0.8769** | **0.8641** | **56.73** | **57.32** | **1.2286** | **1.1797** |
| Drugs (T=500) | GeoDiff | 64.12 | 75.56 | 1.1444 | 1.1296 | 43.16 | 42.02 | 1.3806 | 1.3314 |
| | SubgDiff | 74.30 | 77.87 | 1.0003 | 0.9905 | - | - | - | - |
| | GeoDiff+MSGEN | **85.06** | **97.16** | **0.9538** | **0.9478** | **49.28** | **48.20** | **1.3140** | **1.2719** |
| QM9 (T=500) | GeoDiff | 87.80 | 93.66 | 0.3179 | 0.3216 | 46.25 | 45.02 | 0.6173 | 0.5112 |
| | SubgDiff | **89.40** | 94.39 | **0.2543** | 0.2601 | 49.21 | 47.45 | 0.5030 | 0.4724 |
| | GeoDiff+MSGEN | 88.36 | **95.10** | 0.2606 | **0.2332** | **49.54** | **48.39** | **0.4934** | **0.4624** |

Table 12: Few Steps Generation

**Result and Discussion.** As shown in Table 12, we can see that our framework still significantly outperforms baselines. The results demonstrate that the MSGEN framework maintains strong generation performance in a few-step generation. This robustness stems from explicitly modelling structural priors within the MSGEN architecture. In our framework, the first-stage model implicitly learns to capture coarse-grained molecular structure, such as backbone layout and scaffold topology. This intermediate representation serves as a global geometric anchor for subsequent refinement. By conditioning the second-stage model on this guidance, MSGEN is able to focus its capacity on local corrections and fine-scale geometry while maintaining the structural awareness of key substructures.

## G.5 Complexity-Aware Evaluation

We performed a complexity-aware evaluation by stratifying molecules based on the number of atoms and the number of bonds (edges). This analysis examines whether MSGEN's improvements are uniform across all molecules or more pronounced for more complex structures, which are typically more challenging to generate due to greater conformational variability.

**Results and Discussion** As summarized in Table 13, MSGEN achieves better generation quality across all complexity levels, but the gains are significantly larger for more complex molecules. This trend indicates that our framework becomes increasingly beneficial as molecular complexity increases, highlighting one of its core strengths.

Table 13: Performance comparison by number of atoms and edges.

| Num of atoms | Model | COV-R (%) ↑ | MAT-R (Å) ↓ | COV-P (%) ↑ | MAT-P (Å) ↓ |
|--------------|-------|-------------|-------------|-------------|-------------|
| | | Mean Median | Mean Median | Mean Median | Mean Median |
| < 40 | GeoDiff | 98.23 100.00 | 0.6564 0.6615 | 85.67 91.12 | 0.8520 0.8998 |
| | GeoDiff+MSGEN | **98.34** 100.00 | **0.6482 0.6291** | **87.98 95.97** | **0.8143 0.8025** |
| > 40 | GeoDiff | 85.23 96.00 | 0.9207 0.9118 | 53.37 52.31 | 1.2775 1.2407 |
| | GeoDiff+MSGEN | **88.69 96.00** | **0.8826 0.8466** | **60.42 61.97** | **1.2023 1.1270** |

## G.6 Comparison under Equal Total Diffusion Steps

To further evaluate the effectiveness of our framework, we conduct an additional experiment on GEOM-Drugs where the total number of diffusion steps is kept the same across models. Specifically, we compare our two-stage framework against the standard model with the same total steps. We set *the same number of steps for both stages*. The results are shown in Table 14.

| Num of edges | Model | COV-R (%) ↑ Mean Median | MAT-R (Å) ↓ Mean Median | COV-P (%) ↑ Mean Median | MAT-P (Å) ↓ Mean Median |
|---|---|---|---|---|---|
| < 40 | GeoDiff | **98.31** 100.00 | 0.6322 0.6262 | 86.64 94.44 | 0.8263 **0.7719** |
| | GeoDiff+MSGEN | 97.41 100.00 | **0.6319 0.6037** | **89.06 98.08** | **0.7930** 0.7920 |
| 40 – 50 | GeoDiff | 93.62 **98.75** | 0.8232 0.8158 | 69.27 71.33 | 1.0841 1.0562 |
| | GeoDiff+MSGEN | **94.65** 98.57 | **0.7945 0.8102** | **75.69 79.51** | **1.0084 1.0066** |
| > 50 | GeoDiff | 77.91 85.73 | 1.0018 1.0010 | 40.09 36.87 | 1.4410 1.4090 |
| | GeoDiff+MSGEN | **83.39 90.19** | **0.9583 0.9281** | **47.17 43.21** | **1.3690 1.3403** |

Under this setting, although each individual stage in our framework uses fewer diffusion steps, resulting in larger noise increments per step, our method still achieves superior geometric accuracy and coverage. This demonstrates that the coarse-to-fine structure and molecular upsampling introduced in our framework can effectively guide the denoising process even under more aggressive noise conditions.

Table 14: Comparison under equal total diffusion steps on **GEOM-Drugs** ($\delta = 1.25$Å).

| Total Step | Model | COV-R(%) ↑ Mean Median | MAT-R(Å) ↓ Mean Median | COV-P(%) ↑ Mean Median | MAT-P(Å) ↓ Mean Median |
|---|---|---|---|---|---|
| 5000 | GeoDiff | 87.86 97.00 | 0.8686 0.8545 | 60.17 62.21 | 1.1871 1.1412 |
| | GeoDiff+MSGEN | **91.12 98.52** | **0.8324 0.8207** | **63.26 66.69** | **1.1474 1.1021** |
| 2000 | GeoDiff | 87.82 97.13 | 0.8897 0.8696 | 54.35 56.65 | 1.2506 1.1949 |
| | GeoDiff+MSGEN | **90.05 98.00** | **0.8769 0.8641** | **56.73 57.32** | **1.2286 1.1797** |
| 1000 | GeoDiff | 82.96 96.29 | 0.9525 **0.9334** | 48.21 46.03 | 1.3205 1.2824 |
| | GeoDiff+MSGEN | **85.06 97.16** | **0.9438** 0.9378 | **49.28 48.20** | **1.3140 1.2719** |

This experiment highlights the strength of our generative framework in efficiently guiding the denoising process through coarse-to-fine molecular upsampling. The ability to achieve superior results despite fewer steps per stage demonstrates the robustness and flexibility of our method in maintaining geometric consistency under aggressive noise conditions. This property is particularly valuable when modeling larger and more complex molecular systems.

| Baseline | Variant | Avg Time (s) |
|---|---|---|
| GeoDiff | Vanilla | 223.43 |
| | +MSGEN (2-stage) | 335.46 |
| | +MSGEN (3-stage) | 395.78 |
| ET-Flow | Vanilla | 0.4243 |
| | MSGEN (2-stage) | 0.6739 |
| | MSGEN (3-stage) | 0.8154 |
| ConfGF | Vanilla | 42.87 |
| | MSGEN (2-stage) | 75.10 |

Table 15: Average generation time per molecule in framework generalizability experiments.

## G.7 EFFICIENCY ANALYSIS

Our primary focus in this work is to improve structural fidelity and chemical realism in molecular conformer generation. While computational efficiency is not the central focus, we include an analysis here to characterize the trade-off introduced by our framework. This supplementary evaluation provides insights into the practical cost when plugging our framework into different models.

First, we report the average generation time in the framework generalizability experiments in Table 15. This increase is moderate and justifiable: As show in Appendix C.1, the coarse molecular graphs used in earlier stages are structurally simpler. Consequently, each generation step on the coarse graph is substantially faster, which offsets part of the additional cost introduced by the multi-stage design.

Table 16: Statistics of model checkpoints used in GeoDiff+MSGEN.

| Checkpoint | #Parameters | Size (MB) |
|---|---|---|
| Drugs_1.pt | 1,366,810 (1.37M) | 5.21 |
| Drugs_2.pt | 1,597,724 (1.60M) | 6.09 |
| Drugs_3.pt | 2,415,660 (2.42M) | 9.22 |
| **Total (3-stage)** | **5,380,194 (5.38M)** | **20.52** |

Second, under the same number of diffusion steps, MSGEN achieves better structural fidelity within a shorter average generation time compared to the baselines. This demonstrates that our framework not only preserves efficiency but also leverages structural guidance to reach higher-quality results without extra computational burden. However, we note that this efficiency gain comes at the cost of additional storage requirements(see in Table 16), which represent a trade-off between computational efficiency and storage overhead.

### G.8 STEP ALLOCATION ACROSS 2-STAGES.

To investigate the impact of diffusion step allocation between the first stage and second stage, we conduct an ablation study on the Drugs dataset with a fixed total number of 1000 diffusion steps.

| Allocations | COV-R(%) ↑ Mean Median | MAT-R($\AA$) ↓ Mean Median | COV-P(%) ↑ Mean Median | MAT-P($\AA$) ↓ Mean Median |
|---|---|---|---|---|
| 300+700 | 82.24 96.00 | 0.9828 0.9655 | 45.61 44.67 | 1.3541 1.2930 |
| 400+600 | 83.43 95.72 | 0.9715 0.9676 | 47.56 46.49 | 1.3374 1.2816 |
| 500+500 | **85.06 97.16** | **0.9438 0.9378** | **49.28** 48.20 | **1.3140** 1.2719 |
| 600+400 | 83.57 95.95 | 0.9706 0.9643 | 49.09 **49.50** | 1.3193 **1.2559** |
| 700+300 | 83.97 96.85 | 0.9677 0.9488 | 48.79 48.44 | 1.3205 1.2789 |

Table 17: Step Allocation on Drugs ($\delta = 1.25A$)

**Results and Discussion.** The results are summarized in Table 17. The results reveal that a balanced allocation(500+500) achieves the best overall performance across most evaluation metrics, which suggests that equally distributing generation effort between coarse structure modeling and fine-grained refinement leads to the most faithful and diverse conformers. This ablation study indicates the importance of balanced hierarchical generation, where sufficient capacity is allocated to each stage. It also supports our design choice of a two-stage architecture and motivates further exploration of adaptive step allocation strategies.

### G.9 FRAGMENT-LEVEL COARSE GENERATION

We design an ablation study based on fragment-level coarse structure generation to verify the effectiveness of our geometric upsampling strategy C.7.3.

**Fragmentation.** We decompose the molecule $G = (\mathcal{V}, \mathcal{E})$ into $m$ non-overlapping fragments $F_k$, where $F_k = (\mathcal{V}_k, \mathcal{E}_k)$ and $\mathcal{V} = \bigcup_{k=1}^{m} \mathcal{V}_k$, $\mathcal{E} = \bigcup_{k=1}^{m} \mathcal{E}_k$ using Principal Subgraph(PS) (Kong et al., 2022). The process begins by identifying all unique atomic fragments defined in the fragment vocabulary $\mathcal{S}$, which serves as the basis for coarse-grained representation. Following the progressive strategy in PS, the algorithm iteratively merges the neighboring fragments within each molecule. At each iteration, newly merged fragments are ranked by frequency across the dataset, and the most frequent one is added to the vocabulary. This process continues until the vocabulary reaches a predefined size $|\mathcal{S}|$. By adjusting the target vocabulary size, we can control the granularity of the fragmentation: a smaller vocabulary yields finer partitions and thus more detailed coarse-level molecular structures.

Once the fragmentation is completed, each atom $x_a$ is assigned to its corresponding fragment $x_c$, and we construct a mapping matrix $M$ that encodes the relationship between fine-grained atoms and coarse-grained fragments. This mapping is later used for geometric upsampling and coarse-to-fine

structural reconstruction. We use RDkit (Landrum et al., 2013) to set the initial coordinates of each fragment as the average of its constituent atom coordinates.

Table 18: Geometric evaluation on GEOM-Drugs ($\delta = 1.25\mathring{A}$)

| Models | COV-R(%) ↑ Mean Median | MAT-R($\mathring{A}$) ↓ Mean Median | COV-P(%) ↑ Mean Median | MAT-P($\mathring{A}$) ↓ Mean Median |
|---|---|---|---|---|
| GeoDiff | 87.86 97.00 | 0.8686 0.8545 | 60.17 62.21 | 1.1871 1.1412 |
| GeoDiff+MSGEN | **90.43 97.81** | **0.8530 0.8332** | **63.71 65.13** | **1.1515 1.0939** |

**Experiment Setting.** In our experiment, we set the fragment vocabulary size $|\mathcal{S}| = 50$ to preserve more specific and chemically meaningful fragment structures. Then, we decompose the conformation generation process into two stages. In the first stage, the model generates coarse-grained features at the fragment level. In the second stage, we apply a geometric upsampling strategy to transform the fragment-level representation into a fine-grained structural prior. This guidance is then used to condition the generation of the full all-atom 3D conformation, enabling the model to preserve global coherence while recovering chemically plausible local structures.

**Result and Discussion.** As shown in the Table 18, our framework consistently outperforms the baseline model. These results highlight that geometric upsampling serves as a structurally meaningful bridge between coarse and fine representations. By leveraging fragment centroids and molecular topology to initialize the fine-level generation, the model can better preserve geometric consistency. This confirms the effectiveness of our geometric upsampling design in our framework.

## H  LLM USAGE STATEMENT

In accordance with the ICLR 2026 policy on the use of large language models (LLMs), we report the following statement: LLMs were used solely as writing assistants to polish the language, correct grammar, and improve the readability of the manuscript. No part of the research design, data processing, experiment execution, or results involved the use of LLMs. The authors take full responsibility for the entire content of this paper.

# I GENERATION RESULT

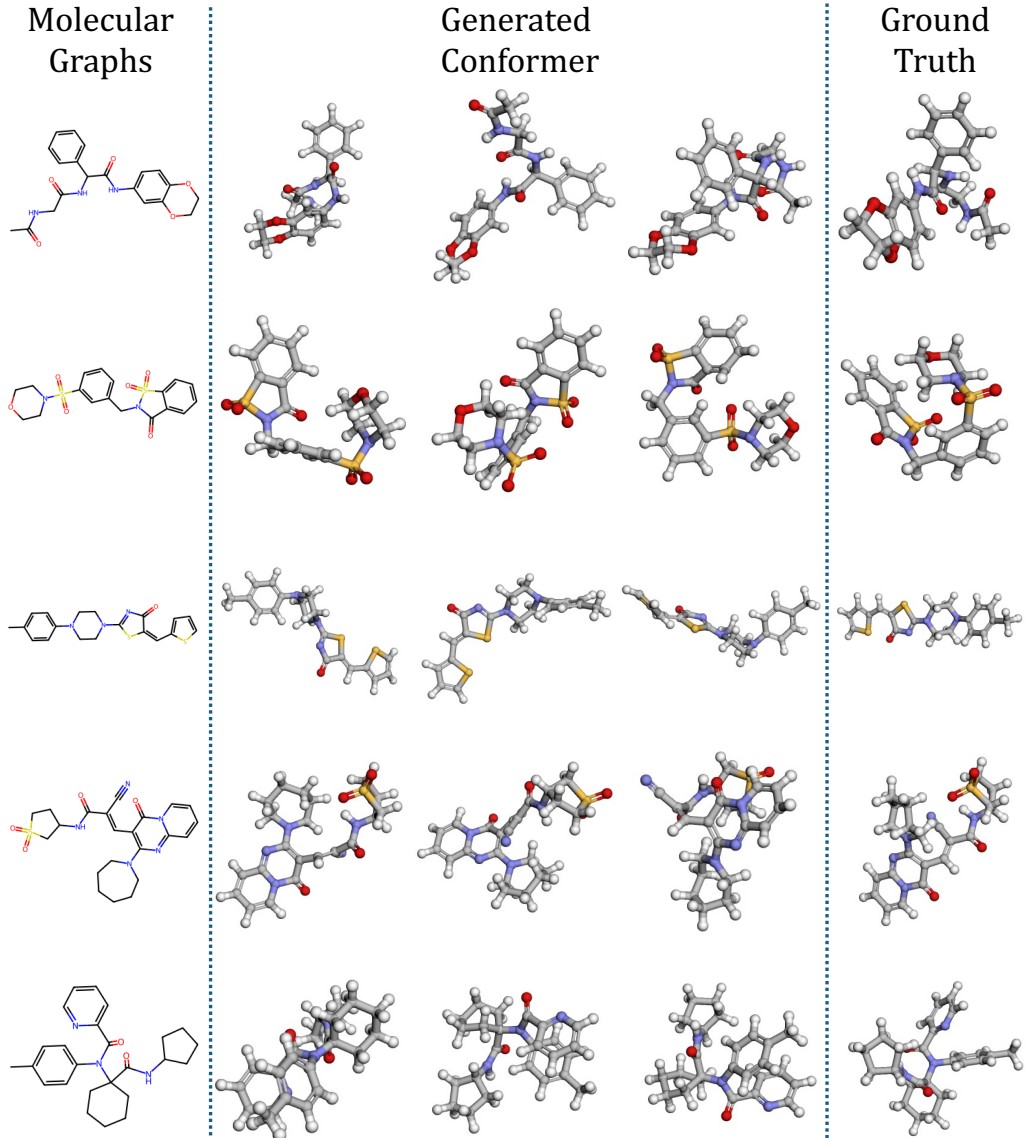

Figure 5: Molecular graphs, generated conformers, and ground truth from GeoDiff+MSGEN (Part 1).

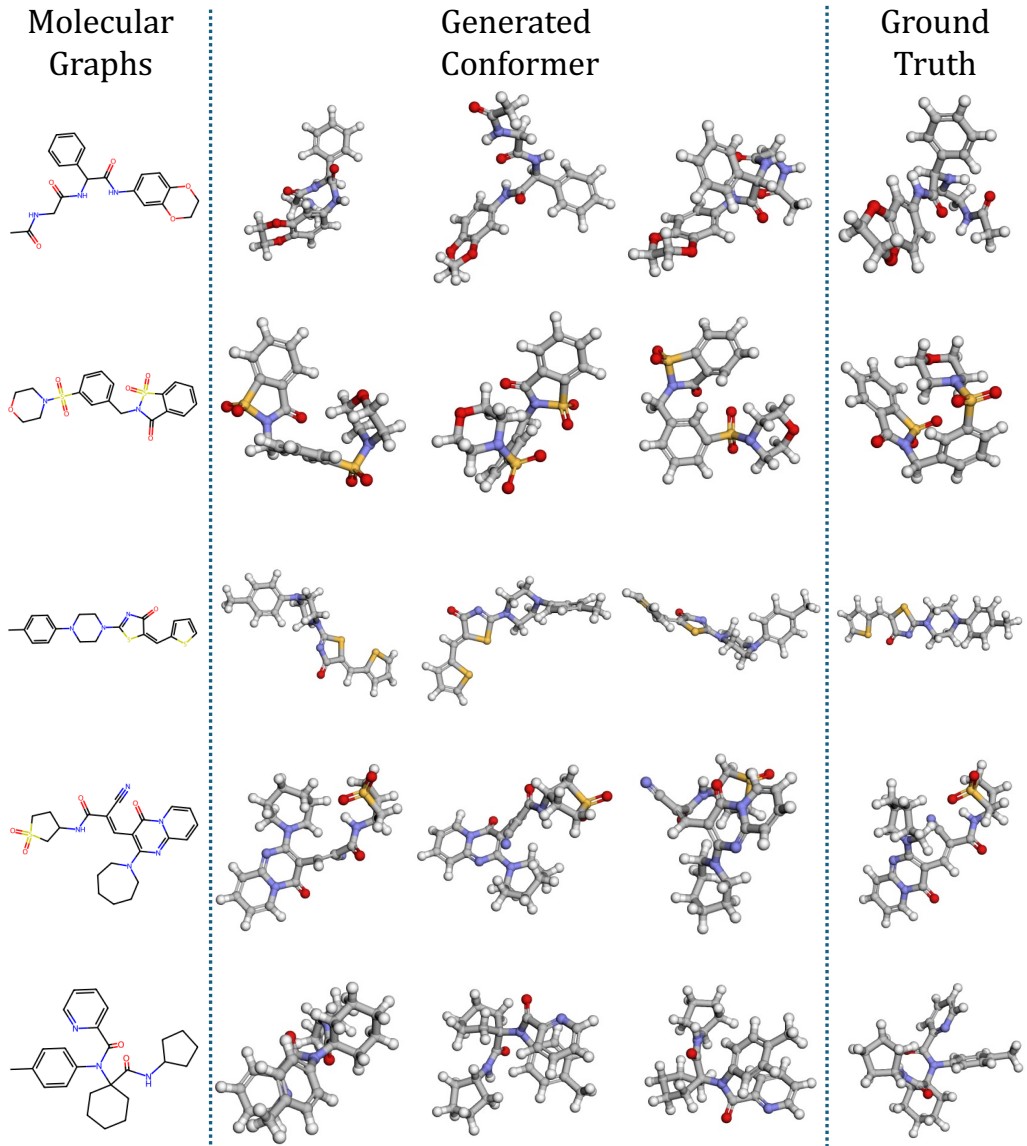

Figure 6: Molecular graphs, generated conformers, and ground truth from GeoDiff+MSGEN (Part 2).

