# OpenReview forum: "Hierarchical Multi-Scale Molecular Conformer Generation"
_ICLR.cc/2026/Conference — ICLR 2026 Poster_

### Official Review · Reviewer_HeM6 · 2025-10-29

**Soundness:** 3
**Presentation:** 3
**Contribution:** 1
**Rating:** 2
**Confidence:** 5

**Summary:**

This paper introduces MSGEN, a hierarchical multi-stage framework for molecular conformer generation. The core idea is to generate molecular structures in a coarse-to-fine manner—first producing coarse substructures such as rigid scaffolds or heavy-atom backbones, and then refining them into full-atom conformations. Each stage is trained independently using a diffusion-based model, and the coarse-level geometry is passed to the next stage via an upsampling-based conditioning module.

The authors claim that this hierarchical design improves structural awareness, stabilizes generation, and produces more chemically valid conformers across standard benchmarks (GEOM-QM9 and GEOM-Drugs). Extensive experiments are presented across multiple backbone models (GeoDiff, ConfGF, ET-Flow, EBD), showing quantitative improvements in coverage (COV) and RMSD (MAT) metrics, as well as minor gains in ensemble property prediction.

Overall, the paper proposes a conceptually clear extension of existing diffusion-based conformer generation frameworks by introducing multi-stage conditioning, with an aim to better preserve molecular substructures and enhance geometric fidelity.

**Strengths:**

- **Clear Motivation and Formulation:**

    The motivation to introduce hierarchical structure into conformer generation is logical and clearly explained. The paper provides an organized mathematical formulation and schematic overview (Fig. 2) that makes the proposed pipeline easy to follow.

- **Compatibility with Existing Diffusion Models:**

    MSGEN is implemented as a general plug-in framework that can be attached to various diffusion or flow-based backbones (e.g., GeoDiff, ConfGF, ET-Flow). This design flexibility increases the practical usability of the method.

- **Comprehensive Experimental Coverage:**

    The authors evaluate their approach on multiple datasets (GEOM-QM9, GEOM-Drugs) and across different base architectures.

- **Empirical Consistency Across Models:**

    The hierarchical conditioning leads to modest but consistent improvements over several backbones, suggesting that the framework is stable and reproducible.

**Weaknesses:**

**Point 1. Ambiguity in Stage-1 Scaffold Arrangement**

The first stage determines the spatial arrangement of isolated scaffolds (Figure 2) without knowledge of the full molecular context (e.g., linker length or flexibility). This arrangement may dominate the final conformation but cannot adapt once the subsequent stages add connecting atoms.

Consequently, the hierarchical formulation appears somewhat unsmooth, and it remains unclear whether this early arrangement can be corrected or whether it meaningfully affects overall generation quality.

**Point 2. Unfair Comparison and Step-Dependent Behavior**

The reported improvements may partially stem from the increased number of diffusion steps rather than from the hierarchical design itself. While the appendix (Table 13) includes equal-total-steps results, the main text does not, preventing fair comparison.

Moreover, GeoDiff + MSGEN with 5 k steps outperforms the 10 k-step version in recall (comparing Table 5 and Table 13), suggesting inconsistent behavior that is neither analyzed nor explained.

**Point 3. Limited Informativeness of the Upsampling Scheme**

The upsampling process simply places new atoms randomly around their topological neighbors. Since these coordinates are only used as conditioning input, it is unclear what meaningful geometric information they (newly added atoms) provide. If the network must relearn local geometry from scratch, the added step contributes little beyond stochastic initialization.

**Point 4. Limited Generality and Lack of Conceptual Novelty**

The hierarchical coarse-to-fine paradigm is well-established in biomolecular modeling. Diffusion-based protein generators (e.g., RFdiffusion, Chroma) already generate backbones first and refine them via side-chain packing or relaxation, while hydrogens are routinely added in post-processing. Therefore, separating hydrogens or rigid scaffolds into multiple stages offers little conceptual advancement. The independence of stage-wise training is also not unique—it parallels conventional coarse-to-fine pipelines widely used in structural biology.

**Point 5. Unconvincing Energy-Based Evaluation**

The task of conformer generation is to recover physically valid ensembles, not to discover novel coordinates. Hence, energetic fidelity is central to evaluation. Yet the proposed method shows larger energy errors than comparable add-on methods (such as Woo et al. [1]), which operates on the same dataset. Although ET-Flow uses a different dataset, its much lower energy errors (0.18 / 0.02 kcal/mol) highlight the expected accuracy. Overall, MSGEN does not convincingly improve energetic realism relative to recent baselines.

[1] Woo, J., Kim, S., Kim, J. H., & Kim, W. Y. (2024). Riemannian Denoising Score Matching for Molecular Structure Optimization with Accurate Energy. arXiv preprint arXiv:2411.19769.

**Questions:**

**Point 1. Scaffold Arrangement Correction:**

How is the relative arrangement of scaffolds in stage 1 adjusted once the full molecular context becomes available? Is there any mechanism allowing later stages to reposition or re-orient scaffolds when linkers are longer or more flexible than expected?

**Point 2. Fair Comparison and Step Allocation:**

Could you provide all main results under the *equal-total-steps* setting? Also, how were the diffusion steps allocated between stages, and why does the 5 k-step version yield higher recall than 10 k?

**Point 3. Information Content of Upsampling:**

What specific geometric information do the upsampled atom positions convey? Have you tested simpler or physically guided alternatives (e.g., random initialization, distance-geometry placement, or short MM relaxation) to quantify their effect?

**Point 4. Novelty Beyond Coarse-to-Fine Tradition:**

In what sense does MSGEN offer conceptual or methodological advances beyond existing hierarchical generation schemes widely used in protein modeling? Is there any plan to automate hierarchy discovery or to demonstrate benefits in domains where coarse-to-fine design is already standard?

**Point 5. Energy-Based Evaluation:**

How were energy metrics in Table 3 computed, and can you include a direct comparison with SOTA models under identical settings?

---

> ### Author Response · Authors · 2025-11-22
>
> **Rebuttal:**
>
> Thank you for recognizing the generality and effectiveness of our method. However, we believe there are some misunderstandings about our paper. We will address your concerns with the following experiments and clarifications.
>
> **Weakness 1 & Question 1:** *Ambiguity in Stage-1 Scaffold Arrangement.*
>
> First, stage 1 already yields higher generation quality than generating from scratch, indicating the initial arrangement is sufficiently good. Second, Stage 2 further refines the results based on Stage 1. We compare the generation quality of coarse-level atoms generated (1) from scratch, (2) after Stage 1, and (3) after Stage 2. Results show Stage 1 provides a substantial improvement, and Stage 2 further boosts the overall quality.
>
> **Table 1:** Geometric Evaluation on coarse-level atoms of GEOM-Drugs in different generation stages
>  |Stage|COV-R(%)↑| MAT-R(Å)↓ | COV-P(%)↑| MAT-P(Å)↓|
> |---|--|---|--|---|
> |from scratch|87.86|0.8686|60.17|1.1871|
> |from stage 1|89.17|0.8543|65.32|1.1293|
> |from stage 2|90.41|0.8424|66.26|1.1217|
>
> **Weakness 2 & Question 2:** *Unfair Comparison and Step-Dependent Behavior.*
>
> To further solve your concerns, our experiments in Appendix G.6 control the generation steps of the two stages for ablation. The results in the following tables show that our method can still outperform the baseline (GeoDiff, ET-Flow) with the same total generation steps. We will move the results to the main paper in a revised version.
>
> **Table 2:** Performance Comparison for Different Baselines under equal total steps.
>
>  |Model|Steps|COV-R(%)↑|MAT-R(Å)↓ | COV-P(%)↑ | MAT-P(Å)↓|
> |--|--|--|--|--|--|
> | ET-Flow|50|74.47|0.5514|55.21|0.7855|
> | ET-Flow+MSGEN |25+25|79.87|0.4673| 62.72|0.7458|
> |GeoDiff|5000|88.16|0.8643|60.57|1.1831|
> |GeoDiff+MSGEN|2500+2500|90.04|0.8441|63.08|1.1523|
>
> For the recall of 10k steps compared to 5k steps, the recall only drops slightly, which does not have statistical significance. Our framework focuses on improving chemical fidelity, better captured by precision metrics, which consistently improve with more steps. This is a reasonable result since the recall metric is close to the upper bound performance, where increasing the number of steps will hardly affect it.
>
> **Table 3:** Performance Comparison for GeoDiff under different total steps.
> |Total Steps |COV-R(%)↑ |MAT-R(Å)↓ |
> |----|----|-----|
> |10000 |90.41|0.8424|
> |6000 |91.15|0.8300|
> |5000 |91.12|0.8324 |
> |4000|90.94|0.8403|
>
> **Weakness 3 & Question 3:** *Limited Information of the Upsampling Scheme.*
>
> We emphasize that the fine-grained atoms, distributed around their chemical neighbors from the first stage, already encode strong geometric information. This is because our upsampling procedure constrains the initialization of these fine-grained atoms to a small region around the coarse-level atoms based on the chemical facts, which is fundamentally different from random initialization.
>
> For your concerns, we present additional results in Appendix C.7, where we experiment with different upsampling strategies: (1) Centroid Upsampling, placing newly added atoms at the geometric center of the coarse structure, and (2) Random Upsampling, which initializes new atom positions from standard Gaussian noise. The results show that our molecular upsampling strategy significantly outperforms random initialization.
>
> **Table 4:** Comparison of MAT for different upsampling strategies
> |Method | MAT-R (Å)↓| MAT-P(Å)↓|
> |----|----|----|
> |Molecular |0.8410|1.1147|
> |Centroid|0.8623|1.1456|
> |Random |0.8791|1.1475|
>
> **Weakness 4 & Question 4:** *Limited Generality and Lack of Conceptual Novelty.*
>
> We believe our method is general due to its hierarchical nature, as noted in your concern: "The hierarchical coarse-to-fine paradigm is well-established in biomolecular modeling." Our method is the first to explicitly incorporate chemical priors into molecular generation, showing its novelty. While the concept is widely used in biology and chemistry, it has not been adopted as a prior in deep learning. This concern highlights the value of our work.
>
> Specifically, existing methods rarely leverage hierarchical structures beyond simple two-stage heuristics. In contrast, MSGEN introduces a coarse-to-fine formulation that aligns with the multi-level organization of molecules, applicable across datasets. Unlike protein pipelines that rely on physics-based relaxation, our method uses a task-agnostic molecular upsampling mechanism based solely on topology, which empirically improves generation quality.
>
> **Weakness 5 & Question 5:** *Unconvincing Energy-Based Evaluation.*
>
> **Response:** We want to emphasize that our framework can reduce the energy compared to the baselines, as shown in Table 3 of the main paper. The provided comparison is not fair, as our paper is orthogonal to the reference. Thus, our method can be applied to it and further reduce energy. However, since the reference code is not open-source, we have not incorporated it into our comparison.

---

> > ### Author Response · Authors · 2025-11-30
> > **Request to Initiate Discussion on Rebuttal**
> >
> > Dear Reviewer HeM6
> >
> > We would like to kindly follow up regarding the rebuttal to your review of our submission. Since we have not yet received a response, we would like to initiate a discussion to clarify any remaining concerns or questions you may have.
> >
> > Best,
> >
> > Authors

---

### Official Review · Reviewer_P6Zq · 2025-10-30

**Soundness:** 3
**Presentation:** 3
**Contribution:** 3
**Rating:** 6
**Confidence:** 4

**Summary:**

This work introduces MSGEN, a hierarchical multi-stage diffusion framework for molecular conformer generation that explicitly models chemical geometry across multiple spatial resolutions. MSGEN decomposes the molecular generation process into a coarse-grained diffusion stage and a fine-grained diffusion stage conditioned on the former, allowing for better-informed generation conditioned on structural geometry. Empirical evaluations demonstrate that this hierarchical approach significantly improves geometric accuracy, structural validity, and diversity over single-scale baselines, and that MSGEN represents a flexible extension for improving existing diffusion models.

**Strengths:**

1. The work additionally provides a motivating study evaluating the performance gains achieved via geometric structural conditioning in conformer generation.
2. The approach outperforms all baselines on conformer generation metrics, and the authors further demonstrate that the method outperforms GeoDiff alone using the same number of total diffusion steps, positioning MSGEN as a flexible extension for existing diffusion-based conformer generators. GeoDiff + MSGEN is also faster at sampling time than GeoDiff alone.
3. The authors justify novel developments and implementation details, including conditional augmentation, molecular upsampling, and 2-stage step allocation with rigorous ablations.

**Weaknesses:**

1. The model requires different parameter sets for each stage in the molecular generation, meaning that model size scales linearly with the number of stages. For a more meaningful comparison, it may be best to scale the benchmark models to a proportional parameter count increase.
2. Regarding the 2-stage framework adopted in the work's main experimental results, the use of anchor-based upsampling to place hydrogens given the heavy-atom backbone seems somewhat unnecessary. Wouldn't inferring their positions instead using any generic toolkit (e.g. RDKit, and perhaps adding some Gaussian perturbation) provide a stronger prior for the second stage?

**Questions:**

1. In table 5, are all methods also performed with the same total number of steps?
2. Would it be possible to train on all stages with a shared backbone that receives the stage $k$ as additional conditioning information? Would this hurt performance significantly?
3. The structural conditioning enabled by a 2-step generation process, starting with the backbone, seems to play a role somewhat similar to self-conditioning (see reference for implementation example), where the model conditions on its own ground-truth predictions. Do any of the baselines implement a similar ground-truth prediction conditioning mechanism? If not, it may be worthwhile to compare MSGEN to a baseline equipped with it.

Watson, J.L., Juergens, D., Bennett, N.R. et al. De novo design of protein structure and function with RFdiffusion. Nature 620, 1089–1100 (2023). https://doi.org/10.1038/s41586-023-06415-8

---

> ### Author Response · Authors · 2025-11-22
>
> **Rebuttal**
>
> Thank you for recognizing the novelty, effectiveness of our methods, and the clarity of our writing. Below, we will solve your concerns with the following answers and experiments.
>
>
> **Weakness 1 & Question 1:** *Need further clarify the comparison to baselines.*
>
> **Response:** We want to emphasize that the target of our work is to improve structural fidelity and chemical realism in molecular conformer generation, where it is usually acceptable to have some trade-offs on efficiency, as shown in existing works in top conferences[1]. In Table 5, the method with our framework will have more steps during generation. To further solve your concerns, we control the total generation steps and the total model size of two stages for ablation based on the experiments in Appendix G.6. The results show that our method can still outperform the baseline (GeoDiff) with the same model size.
>
>
> **Table 1:** Performance Comparison under equal total step and model size
> | Total Step | Model          | Parameters | COV-R(%)↑ | MAT-R(Å)↓ | COV-P(%)↑ | MAT-P(Å)↓ |
> |------------|----------------|------------|------------|-------------|------------|-------------|
> | 5000       | GeoDiff        | 262M         | 88.16      | 0.8643      | 60.57      | 1.1831      |
> | 5000       | GeoDiff+MSGEN  | 268M(106M+162M)        | 90.04      | 0.8441      | 63.08       | 1.1523      |
> | 2000       | GeoDiff        | 262M          | 87.93      | 0.8830      | 54.96      | 1.2456      |
> | 2000       | GeoDiff+MSGEN  | 268M           | 89.14      | 0.8799      | 56.53      | 1.2300      |
> | 1000       | GeoDiff        | 262M           | 83.67      | 0.9500      | 48.97      | 1.3196      |
> | 1000       | GeoDiff+MSGEN  | 268M           | 84.56      | 0.9483      | 48.94      | 1.3177      |
>
>
> **Weakness 2:** *Why not use RDKit for hydrogen upsampling?*
>
> **Response:** We want to emphasize that our framework is more flexible compared to RDKit. Our framework is general to any hierarchy design and can be extended to multiple stages according to the task, but RDKit is restricted to H atoms in two-stage cases. In our 3-stage setting, we use MurkoScaffold[2] to remove not only hydrogens but also omit side-chain atoms, meaning RDKit does not have sufficient structural information to infer their 3D geometry.
>
>  To further solve your concern, we compare our framework with a 2-stage variant where RDKit is used as the upsampling tool under the same experimental setting. As shown in Table 2, the performance of our framework is almost the same as using RDKit, and meanwhile, our framework is more general compared to RDKit.
>
>
> **Table 2:** Performance Comparison for different upsampling methods
>  | Model         | Upsampling | COV-R(%)↑ | MAT-R(Å)↓ | COV-P(%)↑ | MAT-P(Å)↓ |
> |----------------|------------|------------|-------------|------------|-------------|
> | GeoDiff+MSGEN  |   Molecular | 90.41      | 0.8424      | 66.26      | 1.1217      |
> | GeoDiff+MSGEN  |    RDkit   |  89.84      | 0.8431      | 67.06      | 1.1153      |
>
>
> **Question 2:** *Possibility of a shared backbone.*
>
> **Response:**  As we understand, the shared backbone can provide parameter efficiency. But using a shared backbone for all stages is not trivial, as it would require carefully designing how multi-scale structural information is transformed and injected into a unified network without degrading the specialization needed at each stage. We will leave the design of this as future work.
>
> To further solve your concern, we conduct experiments to show that our framework can greatly outperform the baseline with the same number of total parameters without a shared backbone design as shown in Table.1.
>
>
> **Question 3:** *The difference between 2-stage MSGEN and self-conditioning.*
>
> **Response:** We want to emphasize that self-conditioning in [3] is a different concept from our structural condition, and it is orthogonal to our framework. Self-conditioning is a specific technique in diffusion models. The model repeatedly feeds its own predicted clean sample back into the same denoising process to improve temporal consistency, which is not a precise estimation. However, our MSGEN uses precise prediction of the coarse-level atoms as guidance to make the generation of the whole molecule better.
>
>
>
> [1] Torsional Diffusion for Molecular Conformer Generation, Jing et al., NeurIPS 2022
>
> [2] Bemis G W, Murcko M A. The properties of known drugs. 1. Molecular frameworks[J]. Journal of medicinal chemistry, 1996, 39(15): 2887-2893.
>
> [3] Watson, J.L., Juergens, D., Bennett, N.R. et al. De novo design of protein structure and function with RFdiffusion. Nature 620, 108-1100 (2023).

---

> > ### Comment · Reviewer_P6Zq · 2025-11-27
> >
> > Thank you for your detailed response.
> >
> > Regarding weakness 2, the point stands that modelling explicit hydrogens using the multi-stage model is somewhat unnecessary, despite the similar performance. The reviewer understands that MSGEN is flexible to any partitioning of the molecular design process, but the deliberate choice to divide the process into heavy-atom and hydrogen placement (explicitly described as the "main experiment") remains puzzling, as the hydrogen-adding step does not require machine learning modelling at all.
> >
> > Despite this, the reviewer notes the improved performance for 3-step generation and believes this is likely sufficient to demonstrate that MSGEN represents an effective improvement over baselines. Additionally, the reviewer acknowledges that the results reported in Table 1 indicate a slight improvement in most metrics even when fixing the parameter count between the two settings.

---

> > > ### Author Response · Authors · 2025-12-01
> > >
> > > **Response to the Comment:**
> > >
> > > Thank you very much for your thoughtful follow-up and for acknowledging the strengths of our work.
> > >
> > > Regarding the modeling of explicit hydrogens, we appreciate your perspective. Our adoption of the heavy-atom → hydrogen decomposition was intended to demonstrate MSGEN's flexibility and to align with standard practices in molecular modeling pipelines. We believe that this approach offers a practical, efficient, and easily integrable design for modeling. Our experiments focused on demonstrating MSGEN's effectiveness and adaptability to different structural granularities.
> > >
> > > We are also grateful that you recognize the improvement demonstrated in the 3-step setting and the performance gains shown under equal parameter counts in Table 1. If any additional concerns arise, we would be more than happy to address them.
> > >
> > > Thank you again for your constructive feedback and supportive evaluation.

---

### Official Review · Reviewer_iJvk · 2025-10-31

**Soundness:** 3
**Presentation:** 3
**Contribution:** 3
**Rating:** 6
**Confidence:** 3

**Summary:**

This paper introduces MSGEN, a novel hierarchical multi-scale framework for 3D molecular conformer generation. The authors posit that existing deep generative models often fail by overlooking the inherent hierarchical structure of molecules, where key substructures like scaffolds act as anchors for the overall geometry. MSGEN addresses this by generating conformers in a coarse-to-fine process, first generating a coarse-grained structure and then using it as conditional guidance for subsequent, finer-scale stages. The authors demonstrate that this framework can be integrated with a wide range of existing generative models, consistently enhancing their ability to produce more stable, accurate, and chemically plausible conformers, especially for complex drug-like molecules.

**Strengths:**

1. **Chemically-Grounded Motivation:** The paper's premise is strongly rooted in chemical principles. The preliminary study (Section 3) provides excellent justification by showing that a model provided with ground-truth geometric guidance (the heavy-atom backbone) dramatically outperforms other methods, confirming that substructure awareness is critical.

2. **Novel and Necessary Technical Contributions:** The framework introduces two clever solutions to problems specific to this domain, e.g., Molecular Upsampling and Conditional Augmentation, which I find to be novel and reasonable.


3. **8Thorough and Rigorous Experimentation**: The paper is supported by a comprehensive set of evaluations: Geometric and Chemical Evaluation, Generalization, Scalability. The evaluation is not limited to geometric metrics: The authors show that MSGEN-enhanced models produce conformers with lower (better) mean absolute errors on calculated chemical properties like energy and the HOMO-LUMO gap, indicating the generated structures are more physically realistic.

**Weaknesses:**

1. **Potential for Error Propagation:** As with any hierarchical system, errors from the initial coarse stage can be passed on and potentially amplified by the subsequent fine-grained stages. The paper's failure case analysis (Appendix G.3) acknowledges this, suggesting that for highly flexible molecules, small deviations in positional guidance may lead to misaligned ring orientations. Though they argue that this may be raised by GeoDiff, empirical evidence with other backbone models are not provided.

**Questions:**

1. Can you show that MSGEN combined with other backbone models can solve (to some extent) Weakness 1?

2. Did you examine different design choices of molecule upsampling? Current design could be heuristic.

---

> ### Author Response · Authors · 2025-11-22
>
> **Rebuttal**
>
> Thank you for recognizing the novelty, effectiveness of our methods, and the clarity of our writing. We will solve your concerns with the following answers and experiments.
>
> **Weakness 1 & Question 1:**  *Potential error propagation and Further analysis.*
>
> **Response:** We solve your concern, we conduct experiments to measure if a stronger backbone ET-Flow can avoid the error propagation. To be specific, we use ET-Flow to generate molecules that are failure cases in GeoDiff. As shown in the following table, ET-Flow provides better positional guidance on those failure cases of GeoDiff (smaller MAT-R), and thus, it can largely avoid the error propagation (smaller Failure Ratio).
>
>
> **Table 1:** Comparison of MAT-R for failure case Generation
> | Model | MAT-R (Å)↓  | Failure Ratio |
> |------------------- |---------------------|---------------------|
> | GeoDiff            | 1.037               |43/43|
> | ET-Flow            | 0.523               |6/43|
>
>
> **Question 2:** *Different design choices of molecule upsampling.*
>
> **Response:** The principle of our upsampling design is that the fine-level atomic positions should be close to the coarse-level atoms based on the intrinsic molecular graph topology and chemical facts[1,2], where the provided initialization is not heuristic.
>
> To further solve your concern, we have designed other designs of upsampling in Appendix C.7: Centroid Upsampling, which places newly added atoms at the geometric centre of the coarse structure, and Random Upsampling, which initialises fine-level atom positions from standard Gaussian noise. The results in the ablation study and the following table show that our upsampling is better than those two heuristic designs.
>
> **Table 2:** Comparison of MAT-R for different upsampling strategies
> | Method | MAT-R (Å)↓ | MAT-P(Å)↓ |
> |------------------- |---------------------|---------------------|
> | Molecular            | 0.8410               |1.1147 |
> | Centroid            | 0.8623               |1.1456|
> | Random            | 0.8791              |1.1475|
>
> [1] Bemis G W, Murcko M A. The properties of known drugs. 1. Molecular frameworks[J]. Journal of medicinal chemistry, 1996, 39(15): 2887-2893.
>
> [2] Eliel E L, Wilen S H. Stereochemistry of organic compounds[M]. John Wiley & Sons, 1994.

---

> > ### Comment · Reviewer_iJvk · 2025-11-27
> >
> > Thanks for the response! I have no further concerns and will keep my ratings towards acceptance.

---

> > > ### Author Response · Authors · 2025-11-30
> > >
> > > **Response to the Comment:**
> > >
> > > Thank you very much for your positive feedback and supportive decision. We sincerely appreciate the time and effort you have taken to review our work. If you have any further questions or concerns, we would be glad to address them.

---

### Official Review · Reviewer_6BnT · 2025-11-01

**Soundness:** 3
**Presentation:** 3
**Contribution:** 3
**Rating:** 6
**Confidence:** 2

**Summary:**

This work propose a hierarchy molecule generation method. It first generate scaffold, than intermediate structure, and final full molecules. Each stages generations is upsampled and guide the generation of the next stage. Experiments and ablation studies on drug dataset reveals the effectiveness of the method.

**Strengths:**

1. Clear illustration of the method. Figure 1 shows 3 stages in the hierarchy. Figure 2 shows generation process in each stage and upsampling process between stages. Notations are also clearly defined in Section 3, 4.
2. Strong experimental results. Table 2,3,4 shows significantly better score than previous generative models. Table 5 further shows that the hierarchical generation strategy can improve performance of different basemodels.

**Weaknesses:**

1. Missing related work. Similar to this work, which split the whole generation process into different stages, [1] also propose a hierarchical method. It first generate global representation, and then use the global representation as guidance to generate the full molecule. It also achieves good performance on Drug dataset. Therefore, it is necessary to compare hierarchy design and experimental performane between this work and [1].
2. 3 stages needs 3 times inference time and parameters compared to vanilla 1 stage model with the same model size and diffusion inference strategy. Is the comparison with baseline conducted under fair setting? I think the experiments should keep the whole model size, total training time (controlled by training steps), and total inference time (controlled by diffusion steps) similar to the strongest baseline.


[1] Zian Li, Cai Zhou, Xiyuan Wang, Xingang Peng, Muhan Zhang, Geometric Representation Condition Improves Equivariant Molecule Generation, ICML 2025.

**Questions:**

1. The three stage design are largely by heuristic rather than theoretical justification. Therefore, more empirical results should be provided to justify the reason for the hierarchy. Can we design finer hierarchy with more stages? Is there any alternative hierarchy tried?

---

> ### Author Response · Authors · 2025-11-22
>
> **Rebuttal**
>
> We thank the reviewer for highlighting important points regarding the clear illustration and strong experimental result. Below, we address your specific questions and concerns.
>
> **Weakness 1:** *Missing comparison with related work.*
>
>  **Response:** Since GeoRCG[1] focuses on de novo molecular generation, which is different from molecular conformer generation, the experiments between them are not comparable. Compared to GeoRCG, our method has two advantages from the methodology perspective. First, our method does not rely on a pre-trained latent graph encoder, but uses a chemical principle as priors with a clear motivation. Second, our method is more general compared to GeoRCG. We will discuss the comparison in a revised version.
>
> (1) GeoRCG relies on a pretrained latent graph encoder, which requires more effort and is dependent on the training task of the encoder. On the contrary, our method uses explicit chemical multi-scale geometric guidance, whose effectiveness is verified in our preliminary studies. Our method comes from a clear motivation and avoids the black-box nature of the pretrained encoder.
>
> (2) Our MSGEN is designed as a flexible framework that naturally supports additional stages, enabling different integration of structural priors in molecules. However, GeoRCG is hard to extend the stages due to the usage of the pretrained encoder.
>
>
> **Weakness 2**: *Whether comparisons to baseline were conducted under fully fair settings.*
>
> **Response:**  We want to emphasize that the target of our work is to improve structural fidelity and chemical realism in molecular conformer generation, where it is usually acceptable to have some trade-offs on efficiency, as shown in existing works in top conferences[2]. To further solve your concerns, we control the total generation steps and the total model size of two stages for ablation based on the experiments in Appendix G.6. The results show that our method can still outperform the baseline (GeoDiff) with the same generation steps and model size.
>
>
> **Table 1:** Performance Comparison under equal total step and model size
> | Total Step | Model          | Parameters | COV-R(%)↑ | MAT-R(Å)↓ | COV-P(%)↑ | MAT-P(Å)↓ |
> |------------|----------------|------------|------------|-------------|------------|-------------|
> | 5000       | GeoDiff        | 262M         | 88.16      | 0.8643      | 60.57      | 1.1831      |
> | 5000       | GeoDiff+MSGEN  | 268M(106M+162M)        | 90.04      | 0.8441      | 63.08      | 1.1523      |
> | 2000       | GeoDiff        | 262M          | 87.93      | 0.8830      | 54.96      | 1.2456      |
> | 2000       | GeoDiff+MSGEN  | 268M           | 89.14      | 0.8799      | 56.53      | 1.2300      |
> | 1000       | GeoDiff        | 262M           | 83.67      | 0.9500      | 48.97      | 1.3196      |
> | 1000       | GeoDiff+MSGEN  | 268M           | 84.56      | 0.9483      | 48.94      | 1.3177      |
>
>
> **Question 1:** *Heuristic design of Multi-stage and needs for design guidance*
>
> **Response:** The three-stage design in our paper follows chemistry knowledge[3] and is not heuristic. We want to emphasize that the design of multi-stages is dependent on the task and domain knowledge of chemistry. Instead, the focus of our work is to verify the effectiveness of our method in multi-stages. In Appendix E, we provide other possible hierarchy designs, such as using BRICS[4], and the implementation of those is our future work.
>
>
> [1] Zian Li, Cai Zhou, Xiyuan Wang, Xingang Peng, Muhan Zhang, Geometric Representation Condition Improves Equivariant Molecule Generation, ICML 2025.
>
> [2] Torsional Diffusion for Molecular Conformer Generation ,Jing et al., NeurIPS 2022
>
> [3] Bemis G W, Murcko M A. The properties of known drugs. 1. Molecular frameworks[J]. Journal of medicinal chemistry, 1996, 39(15): 2887-2893.
>
> [4] Jorg Degen, Christof Wegscheid-Gerlach, Andrea Zaliani, and Matthias Rarey. On the art of compiling and using’drug-like’chemical fragment spaces. ChemMedChem, 3(10):1503, 2008.

---

> ### Author Response · Authors · 2025-11-30
> **Request to Initiate Discussion on Rebuttal**
>
> Dear Reviewer 6BnT,
>
> We would like to kindly follow up regarding the rebuttal to your review of our submission. Since we have not yet received a response, we would like to initiate a discussion to clarify any remaining concerns or questions you may have.
>
> Best,
>
> Authors

---

### Author Response · Authors · 2025-12-02
**Summary Content To AC**

We thank the AC for reviewing our paper. We would like to highlight a few submission details. Our paper, *Hierarchical Multi-Scale Molecular Conformer Generation with Structural Awareness*, received three positive and one negative reviews.
During the rebuttal, our responses addressed the reviewers’ concerns, and the feedback indicates that these issues have been resolved. The only negative rating is from Reviewer HeM6, who was unable to engage further due to an information-leakage issue. Below, we clarify each reviewer’s primary concerns and explain how we addressed them.

**Reviewer 6BnT: Rating 6, Confidence 2.** The reviewer appreciated the clarity of the writing and the strong empirical performance of our paper. For the weaknesses, the reviewer first asked us to compare our work with existing works and then requested results under the same inference cost settings. In response, we identified the key differences between our setup and the referenced literature, and we additionally ran experiments using the exact settings requested. The resulting comparisons show that our method still outperforms the baselines.

**Reviewer iJvk: Rating 6, Confidence 3.** The reviewer appreciated the motivation, novelty, and effectiveness of our method. Their primary concern was whether using a stronger backbone could address some of our method’s failure cases. In the rebuttal, we provided additional experiments showing that a stronger backbone indeed mitigates these failures. This comparison further highlights the broader generalizability of our framework to different backbones. The reviewer indicated that they had no further concerns.

**Reviewer P6Zq: Rating 6, Confidence 4.**  The reviewer appreciated the clarity of the writing and the strong empirical performance of our paper. For the weaknesses, the reviewer requested (1) experiments under matched inference-cost settings and (2) an ablation study on upsampling. In the rebuttal, we provided additional results demonstrating that our method outperforms the baselines at the same inference cost. We also demonstrated that our approach is more flexible than physical upsampling, while matching the physical method in a special-case setting. The reviewer indicated that our experiments solved the concerns.

**Reviewer P6Zq: Rating 2, Confidence 5.**  The reviewer appreciated the motivation, generality, and effectiveness of our method. However, we believe there may have been some misunderstandings regarding the key aspects of our work, particularly in relation to questions 1, 4, and 5.  Their main concerns were the need for additional evaluation to further validate effectiveness, and questions about the novelty and impact of certain components (e.g., upsampling). In response, we added more evaluations under diverse settings, which show that the anticipated weaknesses do not manifest in practice. We also reiterated the conceptual novelty of our approach and clarified how our components differ from prior work.

We believe that the detailed responses and clarifications we provided have effectively addressed these concerns. However, the reviewer was unable to follow up due to the information-leakage issue. Therefore, we hope AC will carefully consider this matter.

In brief, we are confident that our paper is solid and that we have effectively addressed the reviewers’ concerns, as reflected in the majority of positive ratings and the feedback we received. We also thank the AC again for considering our submission.

---

### Meta-Review · Area_Chair_TaVm · 2026-01-08

**Summary:**

The reviewer HeM6 mainly concerned about the ambiguity of some details.

**Reviewer Concerns:**

The concerns of reviewer HeM6 could be largely addressed with additional information in the rebuttal.

**Reviewer Scores:**

The reviewer HeM6 may increase the score.

---

### Decision · Program_Chairs · 2026-01-26

Accept (Poster)